# Blind test comparison of the performance and wake flow between two in-line wind turbines exposed to different turbulent inflow conditions

Jan Bartl[1], Lars Sætran[1]

[1] Department of Energy and Process Engineering, Norwegian University of Science and Technology, Trondheim, N-7491, Norway

*Correspondence to*: Jan Bartl (jan.bartl@ntnu.no)

**Abstract.** This is a summary of the results of the fourth blind test workshop which was held in Trondheim in October 2015. Herein, computational predictions on the performance of two in-line model wind turbines as well as the mean and turbulent wake flow are compared to experimental data measured at NTNU's wind tunnel. A detailed description of the model geometry, the wind tunnel boundary conditions and the test case specifications was published before the workshop. Expert groups within Computational Fluid Dynamics (CFD) were invited to submit predictions on wind turbine performance and wake flow without knowing the experimental results at the outset. The focus of this blind test comparison is to examine the model turbines' performance and wake development up until nine rotor diameters downstream at three different turbulent inflow conditions. Besides a spatially uniform inflow field of very low turbulence intensity (TI=0.23%) and high turbulence intensity (TI=10.0%), the turbines are exposed to a grid-generated highly turbulent shear flow (TI=10.1%).

Five different research groups contributed with their predictions using a variety of simulation models, ranging from fully resolved Reynolds Averaged Navier Stokes (RANS) models to Large Eddy Simulations (LES). For the three inlet conditions the power and the thrust force of the upstream turbine is predicted fairly well by most models, while the predictions of the downstream turbine's performance show a significantly higher scatter. Comparing the mean velocity profiles in the wake, most models approximate the mean velocity deficit level sufficiently well. However, larger variations between the models for higher downstream positions are observed. The prediction of the turbulence kinetic energy in the wake is observed to be very challenging. Both the LES model and the IDDES (Improved Delayed Detached Eddy Simulation) model, however, are consistently managing to provide fairly accurate predictions of the wake turbulence.

## 1 Introduction

Given the constraints of transmission and installation costs the available area for offshore wind farm installations is fairly limited. Under these circumstances wake interactions play an important role when evaluating the energy production since the energy captured by an upstream wind turbine leaves significantly less energy in the wake for the downstream turbine. For certain wind directions these power losses are estimated to account for up to 10-20% for large offshore wind farms

(Barthelmie et al., 2009). Furthermore, the rotor generated turbulence in the wake is a source for augmented material fatigue on the downstream rotor.

In order to be able to come up with holistic control approaches for optimizing a wind farm well-performing prediction tools for the wake flow behind a wind turbine rotor for all kinds of atmospheric conditions are needed. Therefore, the development of simple wake models began already in the early 1980s. Analytical wake models by Jensen (1983), Ainslie (1988), Crespo et al. (1988), Frandsen et al. (2006) or Larsen et al. (2008) are based on a number of simplifications and calibrated with empirical parameters. Most of the state-of-the-art software used for industrial wind farm planning is still based on these engineering wake models. However, they are not able to reconstruct the wake characteristics to a sufficient degree of detail (Sanderse et al., 2011).

With an increase in computational power advanced CFD models based on more fundamental physics arose. These CFD models are computationally more expensive but are able to resolve the flow structures in much larger detail. In general, two types of CFD approaches are state of the art in wake modelling: Reynolds Averaged Navier Stokes (RANS) equations that are averaging the turbulent fluctuations as well as the computationally more expensive Large Eddy Simulations (LES) which are solving for large eddies only. Hybrid models like Detached Eddy Simulations (DES) combine the advantages of calculating unsteady flow effects from LES as well as resolving small scales in the boundary layers as RANS does. Another challenge is the modelling of the interaction of the wind turbine rotor with the flow: the rotor geometry can either be fully resolved or simplified as a two-dimensional force field. The latter option is usually more efficient with respect to computational time. In RANS models it is possible to fully resolve the rotor geometry and thus model complex three-dimensional flow. In LES models, however, a full resolution of the rotor geometry is difficult as the smaller scales which determine the forces at the surfaces of interaction are not resolved. Thus, the rotor is often modelled as a two-dimensional force field which requires detailed knowledge of the lift and drag forces that act under certain inflow conditions.

Even though the wake behind full-scale wind turbine has recently been measured (Kocer et al., 2011), (Kumer et al., 2015), (Trujillo et al., 2016), the unsteady inflow conditions in full-scale experiments make it very difficult to use those data to verify wake prediction models. Therefore, wind tunnel experiments on model turbines under controlled boundary conditions are an appropriate method to verify simulation tools.

Despite the drawbacks of low Reynolds numbers and possible wall blockage effects in model experiments a number of well-defined comparison tests have been conducted. One of the first model scale experiments was the investigation by Talmon (1985). On a small rotor of the diameter of D=0.36m the wake was measured in order to serve as a reference experiment for calculations. In addition to uniform inflow the wake development is studied in a simulated atmospheric boundary layer. Another seminal investigation was conducted by Medici and Alfredsson (2006). With three-dimensional wake flow measurements on a D=0.18m model turbine down to x/D=9 they shed light on phenomena like wake rotation, wake deflection in yawed operation and bluff body vortex shedding frequencies from the rotor.

At the Norwegian University of Science and Technology (NTNU) two model turbines of the rotor diameter D=0.90m were extensively investigated. Adaramola and Krogstad (2011) were analyzing the effect of modifying tip speed ratio, blade pitch

angle and yaw angle on a downstream turbine. Eriksen (2016) investigated the three dimensional rotor generated turbulence in the wake of one model turbine in detail. Bartl et al. (2012) were examining the wake behind two model turbines, while special attention to asymmetries and wake rotation was given by Schümann et al. (2013). A recent study by Bartl and Sætran (2016) investigated the interrelation of wake flow and the performance of a downstream turbine for axial-induction based wind farm control methods.

The largest rotor investigated for wake comparison studies was the MEXICO rotor with a diameter of 4.5m (Schepers et al., 2010), in which the rotor performance as well as the wake flow are examined in detail. A second campaign investigating even more effects including span-wise pressure distributions, yaw misalignment and unsteady effects was realized at the large German Dutch Wind Tunnel (DNW). A benchmark comparison of the comprehensive set of measurement data with numerical calculations is found in Schepers et al. (2014).

In 2011 the first blind test work shop on turbine performance and wake development behind one model turbine was organized. The geometry of the model turbine and wind tunnel environment was made available to the public and dedicated research groups were invited to predict the model turbine's performance and the wake development up to x/D = 5.0 rotor diameters downstream. A total of 11 sets of predictions were submitted and reported by Krogstad and Eriksen (2013). This first blind test experiment showed a significant scatter in the performance predictions with a variation of several magnitudes in predictions of turbulent quantities in the wake between the different contributions. Therefore, it was decided to perform another blind test workshop in 2012 increasing the test complexity by adding a second turbine aligned with the upstream turbine. The participants were asked to predict the performance of both turbines as well as the wake behind the downstream turbine. Nine different submissions were received showing clear variations in the quality of the predictions between the different modelling methods (Pierella et al., 2014). For a third blind test workshop held in 2013 the complexity was slightly increased again. The two model wind turbines were positioned with a span-wise offset of half a rotor diameter. The results reported by Krogstad et al. (2014) showed that a LES simulation method proved to simulate this complex flow case fairly well. For the present fourth blind test workshop held in Trondheim in October 2015, the focus was directed on the effect of different turbulent inflow conditions on the performance of an aligned two turbine setup. Test cases of low turbulent uniform inflow, highly turbulent inflow as well as non-uniform highly turbulent shear are investigated. The wake flow behind the upstream turbine is analysed which defines the inflow conditions to the downstream turbine. Five different groups contributed with CFD simulations ranging from RANS, over LES to DES computations. Although a general improvement of the results is observed over the years, this report shows up the strengths and drawbacks of the different modelling methods and underlines the persistent importance of validation of CFD codes with well-defined experimental datasets.

## 2 Methods

### 2.1 Test case description

#### 2.1.1 Wind tunnel

The experimental data of this study are measured in the closed-loop wind tunnel at NTNU in Trondheim. The rectangular
test section of the wind tunnel is 2.71 m broad, 1.81 m high and 11.15 m long. The wind tunnel roof is adjusted for a zero
pressure gradient generating a constant velocity in the entire test section. The wind tunnel inlet speed is controlled by an inlet
contraction which is equipped with static pressure holes at the circumferences at two defined cross sections. The wind tunnel
is driven by a 220kW fan located downstream of the test section being able to generate maximum wind speeds of up to
$U_{max}$=30m/s.

#### 2.1.2 Model turbines, rotor and airfoil characteristics

The model wind turbines have a three-bladed rotor with a diameter of $D_{T1}$=0.944m and $D_{T2}$=0.894m. The small difference in
rotor diameter stems from a slightly different hub geometry of the rigs. Apart from that the blade geometry is exactly the
same. Both turbines rotate in the counter-clockwise direction when observed from an upstream point of view. The rotors are
both driven by a 0.37 kW AC Siemens electric motor and controlled by a Siemens Micromaster 440 frequency inverter. The
motor rotational speed can be varied from about 100 – 3000 rpm while the generated power is burned off by an external load
resistance.

The turbine blades were designed using the NREL S826 airfoil from the root to the tip. The airfoil, as shown in Fig. 1, was
designed at the National Renewable Energy Laboratory (NREL) and a detailed description of the airfoil's characteristics is
given by Somers (2005). Herein, the geometry is specified and the performance characteristics estimated. Lift and drag
coefficients are presented for a range of operating Reynolds numbers ($Re_{C,tip,FS}$=$10^6$) for a full scale turbine, which are one
order of magnitude higher than the Reynolds numbers prevailing in this model experiment ($Re_{C,tip,model}$=$10^5$). In order to be
able to characterize the airfoil's performance also at model scale Reynolds numbers, a number of 2D experiments on airfoil
performance have been conducted. Sarmast and Mikkelsen (2013) performed an experiment on a two dimensional S826
wing section of the chord length $c_L$=0.10m at DTU in Denmark. They observed hysteretic behaviour for $Re_c < 10^5$ which is
assumed to be the cause for Reynolds-dependent behaviour of the inner blade elements of the upstream turbine under design
conditions. Another experimental set of S826 airfoil data is presented by Ostovan et al. (2013) from METU in Turkey. They
investigated lift and drag coefficients from $Re_C$=7.15·$10^4$ to $Re_C$=1.45·$10^5$ on a 2D wing with a chord length of $c_L$=0.20m. No
hysteretic effects for low Reynolds numbers are found in this experiment. A third experimental set of airfoil characteristics
from $Re_C$=7.00·$10^4$ to $Re_C$=6.00·$10^5$ has been measured by Aksnes (2015) on a wing section of $c_L$=0.45m at NTNU, Norway.
Neither in this experiment is any Reynolds-dependent behaviour at low Reynolds numbers found. The measured lift and drag
coefficients of these three experiments are in good agreement in the linear lift region, while in the pre-stall and stall region

significant differences between the three datasets are present. For $Re_C=10^5$ DTU's measurements predict stall already at α≈8°, while in METU's and NTNU's experiment stall kicks in a little later around α≈11°. Furthermore, somewhat higher lift values are measured in NTNU's dataset in the pre-stall region compared to the other datasets. Numerical simulations by Sagmo et al. (2016) as well as Prytz et al. (2016) point out strong 3D flow effects caused by stall cells in the pre-stall and

stall region. This could be a possible cause for varying experimental results in this region.

Both rotors are designed for an optimum tip speed ratio of $\lambda_{T1}=\lambda_{T2}=6.0$. The blades are milled from aluminium and the blade tips are cut straight. More details about the blade geometry like detailed chord and twist data is found in an invitational document by Sætran and Bartl (2015).

In this blind test experiment the model turbines are positioned at the wind tunnel center line. The upstream turbine T1's rotor

plane is located at 2.00D from the test section inlet, which is verified to be far enough away to not affect the reference velocity measurement at the inlet contraction. The downstream turbine T2 is positioned 2.77D, 5.18D respectively 9.00D downstream of the upstream turbine rotor. The hub height of both turbines is adjusted to $h_{hub}=0.817$ m. In Fig. 2 a side cut of the wind tunnel is shown indicating a reference coordinate system and the wind turbine positions.

**2.1.3 Inflow conditions**

For this blind test experiment three different turbulent inflow conditions are investigated. This is supposed to shed light on the effects of various turbulence levels as well as shear in the atmosphere on the performance of a wind turbine and its wake. As it is almost impossible to create realistic conditions that resemble atmospheric stability classes in a wind tunnel environment, simplified cases of turbulent inflow are created.

The first inflow condition investigated is a uniform inflow of very low turbulence and is from here on described as Test case A. As shown in Fig. 3 (a) there is no grid installed at the inlet of the test section resulting in a clean and uniform flow. Hot wire measurements at the upstream turbine position give a turbulence intensity level of TI=0.23% of an integral turbulent length scale of $L_{uu}=0.045$m. Over the rotor swept area the mean velocity in the empty tunnel is found to be uniform to within ±0.6%. The boundary layer thickness at wind tunnel walls was measured to be $y_{BL}=0.200$m at the upstream turbine position.

In order to investigate the effects of turbulence on wind turbine performance and wake development, the measurements of Test case B are performed using a large scale turbulence grid at the inlet to the test section (Fig. 3. (b)). The bi-planar grid has a solidity of 35% and is built from wooden bars of 47mm · 47mm cross-section. The grid mesh size is M=0.240m, which generates a turbulence intensity of TI=10.0% at the position of the upstream turbine. The integral length scale here is assessed from an auto-correlation of a hotwire time series is calculated to be $L_{uu}=0.065$m at this position. The grid produces

considerable span-wise variations in the flow, but as soon as the flow reaches the position of the upstream turbine T1 the mean velocity is measured to be uniform to within ±1.5% over the rotor area. Also, the turbulence intensity is assessed to be constant to within ±1.0%. In this grid generated turbulent flow the turbulent kinetic energy is decaying with increasing

distance from the grid. As the flow reaches the first position of the downstream turbine T2, 2.77D downstream of T1, the turbulence intensity in the empty tunnel decays to TI=4.8% while the integral length scale is increasing to $L_{uu}$=0.100m.

In a third Test case C the effect of shear flow combined with high turbulence is investigated. For this purpose a large scale shear flow generating turbulence grid is installed at the inlet of the test section, as shown in Fig. 3 (c). The horizontal mesh width is constant at $M_h$=0.240m while the vertical mesh heights vary between $M_{v,min}$=0.016m near the floor and $M_{v,max}$=0.300m underneath the roof. The grid is bi-planar and has a solidity of 38%. As for the evenly spaced turbulence grid, it is again built from wooden bars of 47mm · 47mm cross-section. At the position of the upstream turbine T1 a turbulence intensity of 10.1% is measured at hub height. The turbulent length scale is estimated to be $L_{uu}$=0.097m for this case. The kinetic energy in the flow is decaying with the distance from the grid. 2.77D further downstream the turbulence intensity has decayed to TI=5.2% while the length scale increases to $L_{uu}$=0.167m. At 5.18D downstream of T1 the turbulence intensity decays to TI=4.1%, while at 9.00D only TI=3.7% remain.

As wind shear and turbulence are generated only at the grid position at the tunnel inlet, their development throughout the tunnel is measured for all turbine positions. Wind shear can be described by the power law in Eq. (1), which expresses the wind speed U as a function of height y, provided that the wind speed at an arbitrary reference height $y_{ref}$ is known:

$$\frac{U}{U_{ref}} = \left(\frac{y}{y_{ref}}\right)^{\propto} \tag{1}$$

The power law coefficient α describes the strength of shear in the wind profile. A wind profile based on a shear coefficient of about α=0.11 is chosen for this experiment resembling the shear at typical stable atmospheric conditions (Hsu et al., 1994), although the grid-generated turbulence in the wind tunnel is much higher than in a stable boundary layer. The mean and turbulent flow profiles at all relevant positions are shown in Fig. 4.

During the present experiments the reference wind speed is kept constant at $U_{ref}$=11.5m/s, which is tested to give a Reynolds-number-independent turbine performance for all inflow conditions. As the downstream turbine T2 experiences significantly lower average wind speeds when operating in the turbulent wake, Reynolds number independent performance characteristics are measured down to an inflow velocity of $U_{inflow}$=6.0m/s at TI=5.0% background turbulence.

For Test case C, in which the velocity is increasing with height, the reference velocity of $U_{ref}$=11.5m/s is set at the turbine hub height $h_{hub}$=0.817m. This reference height is chosen for simplicity reasons; although the rotor-equivalent wind speed (Wagner et al., 2014) that represents the center of kinetic power in the shear inflow is found to be slightly below the turbine hub height (Maal, 2015).

## 2.2 Experimental methods

### 2.2.1 Power and thrust measurements

Both model turbines are equipped with a HBM torque transducer of the type T20W-N/2-Nm, which is connected to the rotor shaft through flexible couplings. In addition to that an optical photo cell is installed on the shaft giving a defined peak signal for every full rotation of the rotor. After subtracting the measured friction in the ball bearing between the rotor and torque sensor, the mechanical power on the rotor shaft can be calculated. The power on both turbines is measured and controlled simultaneously to ensure a stable operation of both turbines.

The thrust force is measured by a 6-component force balance produced by Carl Schenck AG. The drag force on the tower and nacelle structure is first measured without the rotor being present. Thus, it is possible assessing the rotor thrust by subtracting the tower-nacelle drag from the total drag.

### 2.2.2 Wake flow measurements

The mean and turbulent velocities in the wake behind the upstream turbine T1 are measured by a single hot-wire anemometer (HWA) in constant temperature mode (CTA). Each measurement point is sampled for 45s at 20kHz resulting in a total of $9.0 \cdot 10^5$ samples. The signals are amplified and filtered appropriately to avoid the distortion by noise etc. All the wake measurements are repeated using a two-component Laser Doppler Anemometry (LDA) system by Dantec Dynamics for verification. A time series of $5.0 \cdot 10^4$ samples is sampled for a varying period of about 30 seconds. The reference velocity $U_{ref}$ used for normalization of the mean and turbulent wake velocity as well as the non-dimensional power and thrust coefficients is measured at the inlet contraction of the wind tunnel. The pressure difference around the circumferences of two defined cross sections is logged simultaneously for every measuring point. The air density $\rho$ in the experiment is calculated from the measured air temperature and atmospheric pressure in the test section for every measurement point.

### 2.2.3 Statistical measurement uncertainties

The statistical uncertainty of every sample of the power, thrust and mean velocity measurements is calculated following the procedure proposed by Wheeler and Ganji. (2004). Random errors are computed from the standard deviations of the various measured signals on a 95% confidence interval. Taking also systematic errors from the calibration procedures into account by following the procedure of Eriksen (2016), a total error is calculated. Herein, the systematic error of about ±1.0% from the velocity calibration is seen to be the major contributor to the total uncertainty. The uncertainty in the turbulent quantities in the wake flow is calculated according to the approach by Benedict and Gould (1996).

The uncertainty in the upstream turbine power coefficient at design conditions is calculated to be within ±3.0%, while it is lower than ±2.0% for the thrust coefficient. It is observed that the uncertainty of the mean velocity is somewhat larger in the

freestream outside the wake. At higher velocities the sensitivity of the hot-wire probe is smaller which is giving higher uncertainties. The measured values of the turbulent kinetic energy are observed to feature the highest uncertainty in the shear layer between wake and freestream flow.

**2.3 Computational methods**

The computational methods applied by the five different contributors are described in the following sub-sections. Furthermore, an overview of the different simulations methods and parameters is presented in Table 1.

### 2.3.1 Uppsala University and DTU (UU-DTU)

*S. Sarmast*, *R. Mikkelsen* and *S. Ivanell* from Uppsala University, Campus Gotland, Sweden and DTU, Campus Lyngby,
Denmark contributed with a dataset simulated by Large Eddy Simulations (LES) combined with an Actuator Line (ACL) approach. The DTU-in-house code EllipSys3D, which is based on a multiblock finite volume approach, was used to solve the Navier-Stokes computations. The convective terms are herein discretized by a combination of a third order and a fourth order scheme. The resolution of the time domain is defined small enough, that a blade tip moves less than a half cell size per time-step. The flow field around the wind turbine rotor was simulated using the actuator line technique developed by
Sørensen and Shen (2002). Herein, the Navier-Stokes equations are solved with body forces distributed along rotating lines representing the blades of the wind turbine. The lift and drag coefficients are taken from the previously mentioned self-generated dataset for the NREL S826 airfoil by Sarmast and Mikkelsen (2013). For each of the 43 blade points the forces are interpolated for the local Reynolds numbers in a range of 40000 to 120000. Additionally, a force line is introduced account for the drag force generated by the tower. The wake flow field is calculated by solving the Navier-Stokes equations using
LES with an integrated sub-grid scale (SGS) viscosity model.

A regular Cartesian grid which is divided into 875 blocks makes out the computational domain. With 32 points in each block and 43 points representing each blade a total of 28.6 million mesh points is used to simulate the various test cases. This resolution was tested to give a grid-independent simulation result.

The inlet turbulence is modelled by implanting synthetically resolved turbulent fluctuations 1.5D upstream of the position of
the upstream rotor T1. These fluctuations from a pre-generated turbulence field are superimposed to the mean velocities through momentum sources yielding isotropic homogenous turbulence. The mean and turbulent profiles of the different test cases are tested to give a good match with the corresponding wind tunnel values. In addition, the effect of shear flow combined with high turbulence is investigated. The shear profile is implemented to match the profile given in the invitational document by Sætran and Bartl (2015). A more detailed description of the method can be found in Sarmast et al. (2014).

### 2.3.2 Vrije University Brussels (Vrije)

*N. Stergiannis* from Vrije University and Von Karman Institute (VKI) in Brussels, Belgium, performed Reynolds Averaged Navier Stokes (RANS) simulations using the open source software package OpenFOAM in combination with a Multiple Rotating Frame (MRF) approach. Therein, the full rotor geometry is resolved in its own frame of reference and the flow calculated around the "frozen rotor". The subdomain is connected to the stationary frame of reference by an Arbitrary Mesh Interface (AMI). A grid independency test was executed investigating different cell sizes, giving an independent result with a total number of $3.5 \cdot 10^7$ cells. Slip conditions are used at the wind tunnel walls, which was deemed to save computational effort and still takes into account the blockage effect generated by the walls. The rotor and the nacelle are completely resolved, but the turbine towers are not simulated in the final computations. The boundary layers on the blades and nacelle are resolved down to a $y^+ \approx 30$. The standard k-ω turbulence model as implemented in OpenFOAM v.2.4 is applied for the presented simulations. The mean and turbulent inlet velocities were matched with the experimental values provided in the invitational document. As the blade forces could not be directly extracted from the fully resolved rotor simulations, a Blade Element Momentum (BEM) code based on the method by Ning (2014) is used to calculate the power and thrust characteristics of the model wind turbines. The lift and drag coefficients are computed with the open source software XFoil (Drela, 2013) for the NREL S826 airfoil at all prevailing Reynolds numbers. The reference velocity for the downstream turbine is calculated as the average velocity over a line of one radius 1D upstream of the downstream rotor. Only test cases A and B are modelled.

### 2.3.3 Łódź University of Technology (LUT)

*M. Lipian*, *M. Karczewski* and *P. Wiklak* from the Institute of Turbomachinery at Łódź University of Technology, Poland, contributed with two data sets computed by the commercial CFD software ANSYS CFX. All simulations were performed to find a steady state solution of the RANS equations using the k-ω SST model for turbulence closure.

For the test cases A, B and C they fully resolved the rotor geometry. Thus, the solver resolves the actual flow around the rotor and no additional assumptions needed to be made. These simulations will be denoted as Fully Resolved Rotor Model **LUT (FRR)** from now on. Two rotating sub-domains are established around the rotors while the main wind tunnel domain is stationary. A structural mesh is created with the software ICEM CFD to discretize the domains. The wind tunnel is discretized by a total number of $3.0 \cdot 10^4$ plus two refined subdomains around the rotors of $6.0 \cdot 10^3$ nodes each. A grid independence test was executed for the rotor sub domain to prove grid-independent convergence.

For the test cases $B_1$, $B_2$ and $B_3$ a different approach was chosen. The rotors are represented by a custom-made Actuator Line model which will be denoted as **LUT (ACL)**. Herein, the blades are modelled as parallel-epipedons, representing a sub-domain in which the RANS equations are modified. The flow is modified by an addition of force components, which are calculated from tabulated lift and drag data dependent on the local chord and angle of attack. The lift and drag data is taken

from the invitational document and was originally created with XFoil. The ACL model furthermore includes a Prandtl tip loss correction. For these test cases an unstructured mesh is used in the wind tunnel main domain and parallel-epipedon around the blades, discretized by a total number of $1.7 \cdot 10^6$ nodes in the main domain plus two times $1.0 \cdot 10^6$ nodes in the sub-domains around the rotors. As the test cases B and $B_2$ are identical, a direct comparison between the performance and wake results of the FRR and ACL simulations is possible.

### 2.3.4 CD-adapco (CD-adapco)

*S. Evans* and *J. Ryan* from CD-adapco, London, United Kingdom, contributed with a full data set of predictions simulated by Improved Delayed Detached Eddy Simulations (IDDES). The IDDES Spalart Almaras turbulence model is used for turbulence closure in the boundary layers. Both the meshing and the actual simulation is carried out with their commercial software package STAR-CCM+, which is a finite-volume solver using cells of arbitrary shape.

Besides the turbine rotors, the exact geometry of the turbine nacelles, towers and wind tunnel walls is modelled. The computational domain is divided into 3 sub-domains. In the main wind tunnel domain a hexahedral dominant grid is applied, which is further refined around the turbines and in the wake region. In the disc shaped regions around the rotors an isotropic polyhedral mesh of even finer resolution is utilized. The boundary layers around the blade surfaces are resolved down to a $y^+ < 2$. The rotating disk domains around the turbine rotors are connected to the main domain via an arbitrary sliding interface. For the entire computational domain around $2.5 \cdot 10^7$ grid cells are applied.

The inlet conditions are modelled with the Synthetic Eddy Method, generating an inflow field of defined turbulence intensity and length scales that are corresponding to the values given in the invitational document. For test case C, a shear flow is defined by a power law at the wind tunnel inlet. Explicit transient modelling is used to simulate the wind turbine interactions while the turbines' rotations are modelled as a rigid body motion. A transient $2^{nd}$ order model with a time step of $dt=1.0 \cdot 10^{-4}$s is used. Advanced limiter options for minimum limiting and higher order spatial schemes are used in a segregated solver. The transient calculation is run for 1s in test cases A, $B_1$, $B_2$ and C respectively 2.5s in test case $B_3$ due to the higher separation distance. The required values are thereafter averaged for a time period of 0.5s.

More information about the use of Star-CMM+ in rotating flows can e.g. be found in Mendonça et al. (2012).

### 2.3.5 CMR Instrumentation (CMR)

*A. Hallanger* and *I.Ø. Sand* from CMR Instrumentation in Bergen, Norway, provided a data set based on RANS simulations combined with a BEM approach. For the calculation of the mean and turbulent flow quantities their in-house CFD code called Music was used. The RANS equations are solved with a standard k-ε model with Launder-Spalding coefficients. Furthermore, a sub-grid turbulence model is applied to represent the rotor generated turbulence. Therein, it is assumed that the production rate of turbulent kinetic energy and its rate of dissipation are integrated over the wake of the wind turbine and

distributed over the near field. Convective and diffusive fluxes are approximated with the second order Van Leer (1974) and central difference schemes. The turbulent intensity and length scales at the inlet are specified according to the experimental values given in the invitational document for the three different test cases. For test case C, a power law profile is used.

The rotors are included as sub-models in the CFD code. They are represented by their reaction forces on the flow field. The blade forces are simulated by a BEM code including wake rotation. The blades are divided into 30 blade elements in radial direction. The BEM code includes the Prandtl tip-loss correction as well as Glauert's empirical model for highly loaded rotors. The lift and drag coefficients were calculated from the software XFoil (Drela, 2013) in dependence of angle of attack, Reynolds number and relative turbulence intensity. Therein, the transition amplification numbers ($N_{crit}$) are representing the turbulence intensity levels present at the different positions in the wind tunnel. 3D corrections for 2D force coefficients according to the BEM method by Ning (2014) were applied. These forces were used as source terms for axial and rotational momentum conservation. The turbine hubs and towers were modelled as flow resistances in the same control volume as the rotors. Turbine hubs were represented by a drag coefficient of $C_{D,hub}=0.6$, while the tower drag is approximated by $C_{D,tower}=1.2$.

Wind tunnel walls were modelled by wall functions. The entire wind tunnel environment including the two rotors was resolved in a total of $5 \cdot 10^5$ structured grid nodes. Steady state simulations of the blade forces were performed with an angular increment of 15° resulting in a total of 24 azimuthal positions of the turbine rotors. This was deemed to be sufficient to include the effects of shear flow on the first turbine. A detailed description of the computational methods applied is given in Hallanger and Sand (2013).

## 2.4 Required output

In total five different test cases are provided for simulation in this blind test experiment. An overview of the turbines' operating conditions, positioning as well as measurement station of the wake measurements is shown in Table 2.

### 2.4.1 Wind turbine performance

For all five test cases the power coefficients $C_{P,T1}$ and $C_{P,T2}$ (Eq. 2) as well as the thrust coefficients $C_{T,T1}$ and $C_{T,T2}$ (Eq. 3) of both turbines are compared:

$$C_{P,T1/T2} = \frac{8\,P_{T1/T2}}{\rho\,\pi\,D_{T1/T2}^2\,U_{ref}^3}, \tag{2}$$

$$C_{T,T1/T2} = \frac{8\,F_{T1/T2}}{\rho\,\pi\,D_{T1/T2}^2\,U_{ref}^2}. \tag{3}$$

Herein, $P_{T1/T2}$ denotes the mechanical power on the turbine shaft, $F_{T1/T2}$ the thrust force in stream-wise direction on the rotor and ρ the air density. The upstream turbine T1 is operated at a tip speed ratio of $\lambda_{T1}=\omega \cdot D_{T1}/2 \cdot U_{ref}=6.0$, whereas the downstream turbine T2 is run at $\lambda_{T2}=\omega \cdot D_{T2}/2 \cdot U_{ref}=4.5$. Note, that the same reference velocity $U_{ref}$ defined at the test section inlet is used for both turbines. The optimal tip speed ratio for the downstream turbine T2 is also $\lambda_{T2}=\lambda_{T1}=6.0$ when the turbine is unobstructed. As T2 operates in the wake the actually experienced velocity is considerable lower reducing also the optimal rotational speed and thus the tip speed ratio $\lambda_{T2}$. The optimal tip speed ratio at which the maximum power $P_{T2}$ is achieved, in fact varies between $\lambda_{T2}=4.0\text{-}5.0$ depending on the turbine separation distance x/D and inlet turbulence level $TI_{Inlet}$. For better comparability a fixed tip speed ratio of $\lambda_{T2}=4.5$ was chosen.

### 2.4.2 Mean and turbulent wake flow

Furthermore, the horizontal profile of the mean and turbulent flow is compared at the pre-defined wake measurement positions (Table 2). The upstream turbine is still operated at $\lambda_{T1}=6.0$ for all five test cases. The profiles of the normalized mean velocity U* (Eq. 4) and the normalized turbulent kinetic energy k* (Eq. 5) are calculated at the turbine hub height $h_{hub}=0.817$m:

$$U^* = U/U_{ref}, \tag{4}$$

$$k^* = k/U_{ref}^2. \tag{5}$$

In a Cartesian coordinate system the turbulent kinetic energy k is defined as (Eq. 6)

$$k = \tfrac{1}{2}(u_x'^2 + u_y'^2 + u_z'^2). \tag{6}$$

According to Bruun (1995) the single hot-wire (HWA) measures an effective cooling velocity $U_{eff}$ that can be described by the Jørgensen equation (Eq. 7).

$$U_{eff}^2 = U_x^2 + k\,U_y^2 + h\,U_z^2 \tag{7}$$

Dependent on the magnitude of the flow velocity the coefficients k and h typically have values around 1.05 and 0.2 (Bruun, 1995), which means that $U_{eff}$ can be approximated by the velocity perpendicular to the wire. For flows with $U_x \gg U_y$ the effective cooling velocity has the same magnitude as the stream-wise component $U_x$, which is in this case a reasonable assumption for wake measurements at downstream positions starting at x/D=2.77.

Therefore, the isotropic normal stress approximation (Eq. 8) is used to determine the turbulent kinetic energy in each measurement point:

$$k = \tfrac{3}{2} u_x'^2. \tag{8}$$

This approximation is most certainly not appropriate for the zones with high anisotropy, but Krogstad et al. (2014) showed that the isotropic normal stress approximation is a well-fitting approximation in the turbine wake. They measured all three components of the stress tensor with a cross-wire probe for one wake profile at x/D=1 and demonstrated a very good agreement of the isotropic approximation and the component-wise calculation of k.

For the Laser-Doppler (LDA) measurements the stream-wise and cross-wise flow component $U_x$ and $U_z$ are measured. As the stress tensors $u_x$' and $u_z$' from these measurements are seen to be very isotropic, the turbulent kinetic energy k is also in this case approximated by the stream-wise stress $u_x$' only (Eq. 8).

The computed values of mean velocity as well as turbulent kinetic energy from HWA and LDA measurements compare very well. In regions of increased rotation, as in the wake center, the HWA consistently predicts slightly mean velocity lower values. Here, the influence of bi-normal cooling velocity $U_y$ is more pronounced, yet not really significant.

### 2.5 Comparative methods

### 2.5.1 Direct comparison of turbine performances

The predictions of the power coefficients $C_{P,T1}$ and $C_{P,T2}$ as well as the thrust coefficients $C_{T,T1}$ and $C_{T,T2}$ at the pre-defined operating points are directly compared to the experimentally measured values in graphs and tables. The deviations from the measured reference value are discussed on a percentage basis in the text.

### 2.5.2 Statistical performance measures for wake prediction

The predictions of the mean and turbulent wake flow U* and k* are compared in graphs to the measured profiles from the HWA and LDA experiments. In order to provide a more general comparison of the predictions with the experimental results, statistical performance measures are computed as proposed by Chang and Hanna (2004). These measures include the fractional bias (FB), the normalized mean square error (NMSE), the geometric mean bias (MG), the geometric variance (VG) and the correlation coefficient (R). For this purpose, the predictions are compared to the experimental measurements by Hot-wire anemometry (HWA) in the exact same locations as the 41 measurement points along a horizontal line at hub height from z/R=-2.0 to z/R=2.0. Thus, the following statistical performance measures are calculated and compared in tables for each test case:

$$FB = \frac{\overline{x_m} - \overline{x_p}}{0.5(\overline{x_m} + \overline{x_p})}, \tag{9}$$

$$NMSE = \frac{\overline{(x_m - x_p)^2}}{\overline{x_m} \cdot \overline{x_p}}, \tag{10}$$

$$MG = \exp(\overline{\ln x_m} - \overline{\ln x_p}), \tag{11}$$

$$VG = \exp[\overline{(\ln x_m - \ln x_p)^2}], \tag{12}$$

$$R = \frac{\overline{(x_m - \overline{x_m}) \cdot (x_p - \overline{x_p})}}{\sigma_{x_m} \cdot \sigma_{x_p}}. \tag{13}$$

Herein, $x_m$ are the measured values and $x_p$ the predicted values by the models. In this case the compared values x are the normalized mean velocity $U^* = u/U_{ref}$ and normalized turbulent kinetic energy $k^* = k/U_{ref}^2$. The overbar $\bar{x}$ means that an average over all the data points from z/R=-2 to z/R=2 is taken and $\sigma_x$ refers to the standard deviation of the dataset from z/R=-2 to z/R=2.

A perfect model prediction would result in a FB and NMSE=0, and MG, VG, R=1. It has to be stated that these statistical performance measures can by no means give a comprehensive evaluation of the performance of a model, but only provide a general correlation of all data points.

FB and MG are measures of the systematic error, while FB is measured on a linear scale and MG is based on a logarithmic scale. Note that it still might be possible to get a perfect correlation by FB and MG even though the single points are far off at the specific measurement locations. On the other hand, NMSE and VG represent the scatter in the correlation of measured and predicted data and include both systematic and random errors (Chang and Hanna, 2004). Finally, the widely used correlation coefficient R indicates the linear correlation between the measured and predicted value. In this study it is the only measure, which directly compares the predicted and measured value at the specific location. As R is insensitive to addition or multiplication of constants, it is often not recommended as a standalone value for the evaluation of a model (Chang and Hanna, 2004). For the comparison in this blind test experiment, however, the correlation coefficient R is deemed a robust method. The addition or multiplication of the predicted values is in most cases not relevant in the prevailing test cases. All predictions start from the same pre-defined boundary conditions meaning that there is no to big offset in most data.

**3 Results**

The comparisons of the predictions and experimental results are analysed for the different inflow conditions. In chapter 3.1 power, thrust and wake predictions for test case A (low turbulence inflow) are presented. Thereafter, all the test cases for high turbulence inflow conditions for all three separation distances (test cases $B_1$, $B_2$. $B_3$) are analysed in chapter 3.2. Finally, the results of test case C, featuring a highly turbulent shear flow, are compared in chapter 3.3.

Experimental results for power and thrust are indicated by filled black circles for the upstream turbine and empty circles for the downstream turbine. The measurements of the wake profiles with Hot-wire anemometry (HWA) are marked with filled black circles, while flow measurements with Laser-Doppler Anemometry (LDA) are indicated by grey filled circles. The different contributions of numerical simulations are assigned one consistent symbol and colour for power, thrust and wake flow predictions.

### 3.1 Test case A: low turbulence uniform inflow

#### 3.1.1 Power and thrust predictions

The power and thrust predictions for test case A (low turbulence inflow, TI=0.23%) from the five contributions are compared to the experimental results in Fig. 5. The respective numerical values are listed in Table 3.

The experimentally measured power coefficient of the upstream turbine has its maximum $C_{P,max}=0.462$ at $\lambda=6.0$ and its runaway tip speed ratio at $\lambda=11.1$. At a turbine tip speed ratio of about $\lambda=3.5$ a rapid transition of $C_{P,T1}$ into stall is observed. The predictions of the power coefficient of the upstream turbine T1 at its design operating point $\lambda_{T1}=6.0$ show a scatter of about ±7% compared to the measured $C_{P,T1}$. This points out significant differences in the modelling methods. While CMR generated a Reynolds-dependent dataset for lift- and drag coefficients using the airfoil design and analysis code XFoil (Drela, 2013) as an input for their BEM model, UU-DTU used an experimentally generated lift and drag dataset produced by Sarmast and Mikkelsen (2013) as an input for their ACL model. Another aspect is how the predictions modelled the influence of solid wall blockage on the $C_P$ values. As the flow cannot expand freely around the turbine, the induction is reduced, resulting in higher power production of the turbine than that in an unblocked flow. All five contributions took the wind tunnel boundaries into account resulting in fairly well approximations of the upstream turbine's $C_P$ at design conditions.

The scatter in $C_P$ for the downstream turbine T2 is considerably larger than for T1. T2 is operated around its design point at $\lambda_{T2}=4.5$ (referred to $U_{ref}$ measured upstream of T1) in the wake at a separation distance of $x/D_{T2}=5.18$ from the upstream turbine T1. The power is underestimated by up to 25% and overpredicted more than 30% at the most. Some predictions, however, such as CMR, LUT and CD-adapco manage to match the experimental result reasonably well, overestimating the downstream turbine power by only 9-17%, which is a rather small deviation given the large scatter of more than 100% as observed in previous blind test experiments (Pierella et al., 2013), (Krogstad et al., 2014).

The predictions of the thrust coefficient for turbines T1 and T2 give a similar picture, as shown in Fig. 5 (b). Even though the upstream turbine thrust is slightly underpredicted by most simulations, the scatter is significantly smaller than in earlier blind tests. The $C_T$ predictions for the downstream turbine show approximately the same scatter as the upstream turbine. The BEM predictions by CMR matched the experimental results very closely for both turbines.

#### 3.1.2 Wake predictions

For the low inlet turbulence test case A, predictions of the wake flow at $x/D_{T2}=2.77$ behind the upstream turbine are compared. Horizontal profiles of the normalized mean velocity U* and the normalized turbulent kinetic energy k* are compared at hub height as shown in Fig. 6 (a) and (b).

As already observed in a very similar test case in blind test 1 (Krogstad and Eriksen, 2013) the mean velocity profile at x/D=2.77 features two distinct minima located behind the blade tips of the rotor (Fig. 6 (a)). The evident asymmetry in the

wake center is caused by the advection of the tower wake into the swirling rotor wake as shown in rotor wake experiments by Schümann et al. (2013). The wake shape and levels of velocity deficit are very well predicted by CD-adapco and UU-DTU, reflected in well-matching statistical performance measures as presented in the left part of Table 4. Besides small error values of $FB_{U*}$ and $NMSE_{U*}$, the correlation coefficients score of $R_{U*,CD-adapco}=0.960$ respectively $R_{U*,UU-DTU}=0.927$ score significantly better than the other predictions. CD-adapco's IDDES simulations furthermore manage to capture the shape of the wake profile very well, including the asymmetries caused by the tower wake in the center of the profile. Another good prediction of two minima and the correct levels is the fully resolved rotor simulations by LUT. However, the vertical wake extension as modelled by LUT is too small for this low turbulence inflow test case reflected in a somewhat lower correlation coefficient of $R_{U*,LUT}=0.877$. CMR's RANS simulations based on a k-ε turbulence model predict a Gaussian wake shape with only one minimum already at x/D=2.77 downstream of the rotor, suggesting a much more homogenous flow as measured in the experiments. A slightly poorer correlation coefficient of $R_{U*,LUT}=0.877$is therefore calculated. Integrating over CMR's mean wake profile, however, gives a fair estimate of the kinetic energy contained in the wake flow, which is seen in error values $FB_{U*,CMR}$ and $NMSE_{U*,CMR}$ that are approximately zero as well as $MG_{U*,CMR}$ and $VG_{U*,CMR}$ close to the perfect model value one. The reason for that is that these measures do not specifically take the measurement location into account, but are calculated based on different averages over the entire wake. Vrije's method does not resolve the details in the mean velocity profile as the turbine tower was not included in the simulation. The velocity deficit in the wake is significantly underestimated; in average it amounts only about 50% of the experimentally measured values. Still a fairly good correlation coefficient $R_{U*,Vrije}=0.895$ is computed. This unexpectedly high value might be due to the fact that the correlation coefficient is insensitive to addition and multiplication of constants as discussed by Chang and Hanna (2004). This is confirmed by significantly higher deviations of Vrije's prediction in $FB_{U*}$, $NMSE_{U*}$, $MG_{U*}$ and $VG_{U*}$ from the prefect model than the other models.

The normalized turbulent kinetic energy profiles are compared in Fig. 6 (b). The experimental profile shows two distinct peaks in the shear layer generated by the tip vortices around z/R=±1. A third, but substantially smaller peak slightly left to the wake center is ascribed to the turbulence generated by the tower and nacelle structures. It can be observed that the turbulent kinetic energy in the shear layer is very well predicted by UU-DTU's LES as well as CD-adapco's IDDES model, which both match the turbulence peaks generated by the tip vortices perfectly. The statistical performance measures of the turbulence predictions of all models, as presented in the right part of Table 4, a similar picture as previously observed in the mean velocity predictions. CD-adapco is predicting the turbulence profile very well, resulting in a high correlation coefficient of $R_{k*,CD-adapco}=0.938$. The slightly lower correlation of UU-DTU's profile ($R_{k*,UU-DTU}=0.870$) is mainly due to an overprediction of the turbulence generated by the tower in the center of the wake. LUT's RANS simulation based on the k-ω SST turbulence model shows the three distinct peaks, but underpredicts the turbulence levels significantly. This is underlined by considerably higher error values of $FB_{k*,LUT}=0.675$ and $NMSE_{k*,LUT}=0.515$ than in the other simulations. Vrije's simulations based on a k-ω turbulence model indicate the two peaks in the shear layer; but also these predictions give far too low TKE values in the shear layer. In the unaffected freestream flow, however, Vrije's model predicts a significantly too

high TKE, although the freestream turbulence should be pre-defined as an input value. Therefore, a slightly poorer correlation coefficient of $R_{k^*,Vrije}=0.669$ is calculated, while the geometrical variance of the turbulence profile with $VG_{k^*,Vrije}=6.038$ is rather high. CMR's simulation shows two TKE peaks in the shear layer of the same magnitude as in the experimental dataset. However, the turbulence prediction in the wake center and in the freestream are obviously too high, similar as in the aforementioned model. The k-ε model seems not to be able to resolve strong spatial gradients in the distribution of turbulent kinetic energy. Besides a significantly lower correlation coefficient $R_{k^*,CMR}=0.378$ than in the other predictions, the geometrical variance $VG_{k^*,CMR}=89.922$ is almost one order of magnitude higher than in the other predictions.

## 3.2 Test case B: high turbulence uniform inflow

### 3.2.1 Power and thrust predictions

A second set of power and thrust predictions is compared for inflow conditions of higher turbulence. A turbulence grid installed at the wind tunnel inlet is generating a uniform wind field with a turbulence intensity of TI=10.0% at the location of the first turbine rotor. For this high background turbulence level the turbine power and thrust are compared for three turbine separation distances x/D= 2.77, 5.18 and 9.00 (test cases $B_1$, $B_2$ and $B_3$). The power and thrust predictions for test case B are compared in Fig. 7 (a)-(f). A comparison of the respective numerical values is presented in Table 5.

Comparing the upstream turbine power curve for high background turbulence (test cases $B_2$, Fig. 7 (c)) to the upstream turbine power curve of low background turbulence (test case A, Fig. 5 (a)) a very similar curve shape is observed. At increased background turbulence the maximum power coefficient is measured at the same level as for low background turbulence. Also, the runaway tip speed ratio at λ=11.4, at which the rotor does no longer produce energy, is very similar for both inlet turbulence levels. The most noticeable difference is the transition to stall at a tip speed ratio of about λ=3.5 and lower. For higher background turbulence the transition into stall is much smoother compared to low inlet turbulence.

The predictions of $C_{P,T1}$ at its design operating point $λ_{T1}=6.0$ are again very accurate, scattering only about ±7% around the experimental value. Also, the predictions of the thrust coefficient $C_{T,T1}$ are matching very well. As previously observed in test case A, the $C_{T,T1}$ is slightly under predicted, in this case up to -9% at its most. Comparing the performance results of the downstream turbine, the best predictions are made for the lowest turbine separation distance x/D=2.77 (test case $B_1$, Fig. 7 (a)). The experimentally measured power coefficient $C_{P,T2}$ is well matched, with a total deviation of about ±15%. The downstream turbine thrust coefficient $C_{T,T2}$ is predicted within ±10% by all the modellers in test case $B_1$. The predictions by CMR and CD-adapco match the experimental results closest.

Increasing the turbine separation distance to x/D=5.18 in test case $B_2$ the scatter in the results becomes significantly larger (Fig. 7 (c)). The scatter in the downstream turbine power coefficient $C_{P,T2}$ increases to about ±20% in both directions. The fully resolved rotor model (FRR) by LUT results in a very good prediction of the downstream turbine power coefficient, while their actuator line model (ACL) overpredicts the power significantly. This can be directly related to different wake

flow predicted by the two models. The wake flow acts as inflow for the downstream turbine (compare Fig. 8 (a) further down). In contrary, UU-DTU's Ellipsys3D calculation underpredicts the downstream turbine performance significantly, even though the wake characteristics are predicted very accurately. Also Vrije underpredicts the downstream turbine power significantly. This is rather surprising as the wake deficit at x/D=5.18 is slightly underpredicted as well and more power should be left in the flow for the downstream turbine. The scatter in the thrust calculations, as presented in Fig. 7 (d), is in general smaller than for the power predictions for all models, with most simulations underpredicting the experimental value. The thrust coefficient is less sensitive to a correct prediction of the incoming velocity field than the power coefficient. The thrust coefficient is indirectly proportional to the incoming velocity squared ($\sim U_{ref}^2$), while the power coefficient is even more sensible to an incorrect prediction of the incoming velocity field ($\sim U_{ref}^3$). Surprisingly, LUT's FRR model gives the smallest value for the downstream turbine thrust coefficient, although the power and wake predictions for this downstream distance are matching the experimental results very well.

With a further increase in turbine separation distance to x/D=9.00 (test case $B_3$) the experimentally measured downstream turbine power coefficient recovers to $C_{P,T2}$=0.270. The variation in the simulations, as shown in Fig. 7 (e), is seen to be even bigger for this downstream distance reaching a scatter of more than 30%. The same trend as already seen for smaller separation distances is observed: UU-DTU's and Vrije's simulations are clearly underpredicting the power coefficient, while LUT's ACL model is overestimating the downstream turbine power considerably. The thrust predictions show similar tendencies as the power predictions but are seen to match the experimentally measured value better (Fig. 7 (f)).

### 3.2.2 Wake predictions

For the high background turbulence test case B, the participants were asked to predict the mean and turbulent wake characteristics at three downstream distances $x/D_{T2}$=2.77, 5.18 and 8.50. Note that the horizontal wake profiles were extracted from test case $B_3$, in which the downstream turbine T2 was installed at $x/D_{T2}$=9.00 and operating at $\lambda_{T2}$=4.5. The wake flow as measured at $x/D_{T2}$=8.50 is therefore experiencing the induction of the downstream turbine which is located only $x/D_{T2}$=0.50 further downstream. The horizontal wake profiles of the normalized mean velocity $U/U_{ref}$ and normalized turbulent kinetic energy $k^*=k/U_{ref}^2$ are compared in Fig. 8 (a)-(f).

The wake characteristics of the flow $x/D_{T2}$=2.77 downstream of T1 are presented in Fig. 8 (a) and (b). For this case, LUT simulated the wake flow with two different models, the simpler actuator line model (ACL) and the computationally more expensive fully resolved rotor (FRR) model. At this downstream distance the mean wake profiles are characterized by two distinct minima. The experimental results clearly show that a Gaussian wake shape has not yet developed. A very accurate prediction of the mean wake shape is given by UU-DTU's simulation, but also CD-adapco and the FRR model by LTU capture the shape very well. LTU's ACL model, however, only predicts one distinct minimum in the mean wake profile. Only one minimum is also predicted by CMR while the mean velocity profile is rather skewed. Vrije's simulations match the

experimental measurements significantly better for a higher background turbulence level than for the lower turbulence level of Test case A, predicting both the level and wake shape fairly well.

The fact that all predictions approximated the level of mean velocity deficit fairly well is also reflected in the statistical performance measures as presented in Table 6 (upper left section). $FB_{U*}$ and $NMSE_{U*}$ are close to zero, while $MG_{U*}$ and $VG_{U*}$ show only very small deviations from the perfect correlation value one for all predictions. The highest correlation coefficient $R_{U*}$ is reached by CD-adapco with 0.970, closely followed by UU-DTU, Vrije and the FRR model by LUT. The correlation coefficient of CMR's prediction is a few percent lower, while LUT's ACL model that only predicts one minium scores lowest.

Very good predictions of the distribution of the turbulent kinetic energy are presented by CD-adapco as well as UU-DTU. Both simulations predict the magnitude and location of the two peaks around z/R=±1 as well as the region of lower turbulence into the center of the wake very accurately. This is also reflected in the high values of the correlation coefficient $R_{k*,CD-adapco}=0.912$ respectively $R_{k*,UU-DTU}=0.911$ as shown in the upper left section of Table 6. LUT's FRR simulation manages to reproduce the general shape of the turbulence profile, but the levels are about 50% below the measured turbulence values resulting in a significantly lower correlation coefficient $R_{k*,LUT(FRR)}=0.720$. Similar levels are observed for LUT's ACL simulation, which is additionally smearing out the turbulence to the center of the wake giving a correlation coefficient of $R_{k*,LUT(ACL)}=0.468$. It has been discussed that the tip loss correction model included in the ACL model could have contributed to kill the turbulent peaks. Vrije's model based on a standard k-ω turbulence model underpredicts the peaks in the shear layer significantly; they are observed to be lower than the turbulence levels in the freestream flow, which are overpredicted by more than one magnitude. A very low and negative correlation coefficient of $R_{k*,Vrije}=-0.008$ confirms this observation. The negative sign stems from a mainly negative correlation, meaning that turbulence levels are predicted to decrease from the freestream to the shear layer, while they are actually increasing in the experimentally measured profile. CMR's simulations predict too high turbulence levels at the peaks, but surprisingly also in the wake center and in the unaffected freestream flow. A rather low correlation of $R_{k*,CMR}=0.417$ with the experimental data is achieved, while the normalized mean squared error $NMSE_{k*,CMR}=0.698$ is significantly higher than for the other predictions. A possible reason for that blurry turbulence distribution could be the k-ε turbulence model used.

Moving downstream to $x/D_{T2}=5.18$ a more Gaussian mean velocity profile with only one distinct minimum develops as shown in Fig. 8 (c). The general shape of the mean velocity profile is in this case well predicted by almost all the simulations; only Vrije's simulation indicates a near-wake shape with two minima but still results in fairly well statistical performance measures as presented in the middle left section of Table 6. Again, UU-DTU's model is giving a very good match with the experimentally measured profiles, which is also reflected in very low $FB_{U*}$ and $NMSE_{U*}$ values. $MG_{U*}$ and $VG_{U*}$ approach the perfect value one almost very closely and a very high correlation coefficient of $R_{U*,UU-DTU}=0.964$ is calculated. CMR's model computes a slightly asymmetric mean wake profile underpredicting the velocity deficit somewhat, but still is overall performing well as indicated in the correlation coefficient of $R_{U*,CMR}=0.937$. LUT modelled the 5.18D wake using their simpler ACL model, which is underpredicting the mean velocity deficit considerably. The statistical

performance measures are therefore slightly poorer than for the other predictions for this case as shown in Table 6. CD-adapco's IDDES simulation overpredicts the mean wake velocity deficit to some extent, but still reaches the highest correlation coefficient $R_{U*,CD-adapco}=0.971$. This might be due to the almost perfect correlation of the flow in the freestream and shear layer, although the mean velocities in the wake center are predicted somewhat lower than measured in the experiment.

The turbulence profiles for $x/D_{T2}=5.18$ as presented in Fig. 8 (d) show a similar picture as seen earlier for $x/D_{T2}=2.77$. The best predictions are made by CD-adapco's IDDES computation and UU-DTU's LES simulation, with both predictions resulting in very low error indicators $FB_{k*}$ and $NMSE_{k*}$. A very high correlation coefficient $R_{k*,CD-adapco}=0.934$ to the experimental dataset is achieved by CD-adapco's prediction, although the turbulence peaks in the tip vortex region at $z/R=\pm1.0$ are somewhat overpredicted. The magnitude of the peaks in the shear layer is almost perfectly predicted by UU-DTU's computation. Compared to the experimental dataset the peaks are however too broad, overpredicting the TKE in the wake center. This is reflected in a fairly well, but somewhat lower correlation coefficient of $R_{k*,UU-DTU}=0.850$. Too smooth turbulence profiles are predicted by CMR as well as LUT's ACL model, clearly overpredicting ($MG_{k*,CMR}=0.483$) respectively underpredicting ($MG_{k*,LUT}=1.495$) the mean turbulence levels. Vrije's turbulence prediction is very similar to the profile measured at $x/D=2.77$ and resulting again in a rather low correlation coefficient of $R_{k*,Vrije}=0.371$.

A challenging test case is shown for the wake measured at downstream position $x/D_{T2}=8.50$, only half a rotor diameter upstream of the rotor of T2 (Fig. 8 (e) and (f)). A smooth Gaussian mean velocity profile has developed while velocity deficit is further decreasing. Again, UU-DTU is predicting the mean wake well, scoring the highest in the correlation coefficient $R_{U*,UU-DTU}=0.970$ as shown in the lower left section of Table 6. Although, the mean profile predicted from LUT's ACL model is matching the experimental values very well for this case it is very similar to the profile predicted already for 5.18D, where it was clearly underpredicting the velocity deficit. Very low error values of $FB_{U*,LUT}$ and $NMSE_{U*,LUT}$ are computed, while $MG_{U*,LUT}$ and $VG_{U*,LUT}$ are close to one. The correlation coefficient $R_{U*,LUT}=0.936$ is fairly good, but scoring slightly lower than the other predictions. This might be due to obvious discontinuities of the mean velocity profile at $z/R=\pm1.7$. CD-adapco's simulation is strongly overpredicting the mean velocity deficit in the wake at this downstream distance. Surprisingly, the mean velocity deficit even grows noticeably in comparison to the mean wake profile predicted at 5.18D. As shown in the numbers in the lower left section of Table 6, this obvious deviation is also resembled in significantly higher deviations of the mean geometrical bias $MG_{U*}$ and geometrical variance $VG_{U*}$ than the corresponding values of the other predictions. Also Vrije's simulation overpredicts the mean velocity deficit for this case. Correspondingly, $MG_{U*}$ and $VG_{U*}$ give the second highest deviation from the experimentally measured profile. Remarkably, the averaged velocity deficit at 8.50D has not recovered very much from the one predicted at 5.18D. As observed for smaller downstream distances already, CMR predicts a slightly too low velocity deficit also for 8.50D. Almost all statistical performance measures for CMR, however, are significantly better at this far wake distance than at the closer measurement stations.

Analyzing the turbulence profile as shown in Fig. 8 (f) the tip vortex peaks have decayed to about 50% of the magnitude measured at 5.18D. Both CD-adapco's IDDES as well as UU-DTU's LES simulation give a fairly well approximation of the

turbulence profile, as reflected in the highest correlation coefficients $R_{k*,CD-adapco}=0.811$ and $R_{k*,UU\_DTU}=0.812$. As the decay of the turbulence in the wake center is slightly underpredicted by both simulations, these values do not score as high as for the near-wake measurement stations. CMR is overpredicting the turbulence levels at 8.50D, smearing out the turbulence profile to an almost constant line. The acceptably good correlation coefficient $R_{k*,CMR}=0.804$ is giving a wrong impression in this case, as R is insensitive to addition as introductorily stated in chapter 2.5.2 and the profile is basically shifted upwards. The high deviations from 1.00 in $MG_{k*,CMR}$ and $VG_{k*,CMR}$, however, indicate the significant mismatch.

On the other hand LUT's ACL model is underpredicting the turbulence considerably. Higher deviations in $MG_{k*,LUT}$ and $VG_{k*,LUT}$ are observed than for the other predictions. The turbulence levels predicted by Vrije's k-ω model at 8.50D are observed to be very similar to those already predicted at lower separation distances. This indicates that the turbulent decay rate is not well captured for this case. Compared the lower separation distances the predicted TKE profile matches better with measured profile, resulting in acceptable statistical performance measures (e.g. $R_{k*,Vrije}=0.656$).

### 3.3 Test case C: high turbulence non-uniform shear flow

### 3.3.1 Power and thrust predictions

For the last test case the complexity of the inflow conditions is increased. The inflow to the test section is no longer spatially uniform. Another custom-made grid with vertically increasing distance between the horizontal bars is placed at the test section inlet generating a shear flow that can be approximated by the power law exponent α=0.11. The background turbulence of this grid is measured to be TI=10.1% over the rotor area at the location of the first turbine rotor. This makes the effects of shear flow well comparable to test case B as basically the same background turbulence level is predominating. For test case C the turbine power and thrust are compared only for one turbine separation distance $x/D_{T2}=5.18$. The power and thrust predictions for the shear flow test case are presented in Fig. 9 while the exact numerical values are shown in Table 7.

Comparing the upstream turbine power curve of test case C (Fig. 9 (a)) to the upstream turbine power curve of uniform inflow test case B (Fig. 7 (c)) a very similar curve shape is observed. Taking a closer look, however, a slightly lower maximum power coefficient is measured in case C and a marginally earlier run-away point is found at λ=11.2. This is assumed to stem from the fact that the reference velocity $U_{ref}$ for this test case is defined at the center of the rotor at hub height. Due to the vertically non-linear gradient in velocity distribution (see Fig. 4), the rotor equivalent wind speed (Wagner et al., 2014) is found to be slightly higher than $U_{ref}$ measured at hub height. Therefore, the $C_P$ and $C_T$ calculations that are a priori defined to be referred to the hub height reference wind speed $U_{ref}=11.5$ m/s are slightly lower for test case C than for test case B. The rotor swept area is exposed to the same kinetic energy in cases B and C, the wind speed at the predefined reference height in test case C, however, does represent the rotor averaged wind speed (for a more detailed investigation the reader is referred to Wagner et al., 2014).

The predictions of $C_{P,T1}$ at the turbines design operating point $\lambda_{T1}=6.0$ are again very precise, showing a scatter of less than $\pm5\%$ from the measured value. All the contributions predict a little lower $C_{P,T1}$ value as in test case B confirming the tendency measured in the experiment. Also, all the predictions of the thrust coefficient $C_{T,T1}$ give a very good match with the experiment. In this case the spread is about $\pm5\%$ which is just slightly outside the measurement uncertainty.

Analyzing the performance results of the downstream turbine at $x/D_{T2}=5.18$ yet the predictions are very good. The scatter in $C_{P,T2}$ is within $\pm7\%$, except from UU-DTU's prediction that is about 24% lower than the experimental value. This seems to be a systematic deviation as significantly low values have been observed in test cases B already. The predictions of the thrust coefficient are very close to each other, however up to 16% lower than the measured value at $\lambda_{T2}=4.5$. A general tendency in underpredicting the thrust is again seen for all test cases (A, B, C), but the predictions are significantly closer compared to previous blind test comparisons.

### 3.3.2 Wake predictions

One single wake profile behind the upstream turbine is compared for test case C in which the turbine is exposed to highly turbulent shear flow at the test section inlet. The mean and turbulent wake characteristics at $x/D_{T2}=2.77$ behind the upstream turbine are compared in Fig. 10.

The mean velocity profile (Fig. 10 (a)) has a very similar shape as the wake behind the same turbine exposed to uniform inflow of the same turbulence intensity (Fig. 8 (a)). Also the mean velocity profile for shear inflow is characterized by two distinct minima and a smooth transition from the wake to the freestream. Taking a closer look the wake in case C is slightly skewed compared to the one measured in test case B. Especially the minimum velocity peak at $z/R\approx-0.7$ is somewhat lower as in test case B. It is assumed that low kinetic energy fluid that encounters the lower half of the rotor is transported into the measurement plane by the rotation in the wake. Turbulent mixing processes have most likely evened out this effect already at $x/D=2.77$, yet a small difference is detectable.

Four different predictions are compared as Vrije did not simulate test case C. As observed for the earlier test cases, UU-DTU's LES simulation predicts the mean wake shape very accurately. The levels of the two minima are matched very closely, which is also reflected in a high correlation coefficient of $R_{U*,UU-DTU}=0.965$ as presented in Table 8. LUT's fully resolved rotor simulation gives a good agreement as well ($R_{U*,LUT}=0.952$); the skew in the wake is however not as distinct as in the measured profile. CD-adapco predicts the skewed shape of the wake very well as indicated in the highest correlation coefficient $R_{U*,CD-adapco}=0.972$ for this test case; the kinetic energy deficit however is again slightly too high in the blade tip regions, which is reflected by slightly higher deviations in the fractional bias $FB_{U*}$ and geometrical mean bias $MG_{U*}$. As previously observed for test case B the two mean velocity minima are melted into only one in CMR's simulations. Nevertheless, the simulations predict skew in the mean wake profile when comparing to CMR's mean wake prediction for test case B. The correlation coefficient $R_{U*,CMR}=0.898$ is therefore slightly lower than for the other predictions, but indicates an overall well performance.

Analyzing the turbulent kinetic energy profiles for test case C (Fig. 10 (b)) obvious similarities to the ones of test case B (Fig. 8 (b)) are observed. UU-DTU's simulations match the experimental results very accurately in the center and the tip region, whereas the turbulence level in the freestream is slightly too high. A similar correlation coefficient $R_{k*,UU-DTU}=0.866$ as for test case B is computed. LUT's FRR simulations underpredict one peak significantly while the turbulence level in the freestream is significantly higher than in the measurements. This is also reflected in a poorer correlation with the experimental data, as a correlation coefficient of $R_{k*,LUT}=0.666$ is achieved. The TKE predictions by CD-adapco give very close match to the experimental values for this case. The turbulence peaks in the shear layer as well as the freestream level match the measured profiles very well, while the levels in the wake center are insignificantly underpredicted. The resulting correlation coefficient $R_{k*,CD-adapco}=0.795$ is almost the same magnitude as $R_{k*,UU-DTU}$. Similar observations as in test case B are made for the turbulence predictions of CMR. Although the shear layer peaks are on the same level as the experimental values, the levels of turbulence in the wake center and the freestream flow are significantly overpredicted. This observation is confirmed by significantly poorer $MG_{k*}$ and $VG_{k*}$ than for the other predictions as shown in the right section of Table 8.

## 4 Discussion and conclusions

Five different research groups predicted the performance and wake flow between two in-line model wind turbines with a number of different simulation methods. The methods cover different approaches, ranging from commercial software to in-house developed codes. The effect of three different inflow conditions, low turbulence uniform inflow (A), high turbulence uniform inflow (B) and high turbulence non-uniform shear inflow (C) is investigated.

The performance of the upstream turbine ($C_{P,T1}$, $C_{T,T1}$) was commonly predicted rather well by all predictions for all three inlet conditions, with an acceptable scatter of ±5% to ±7% depending on the test case. The upstream turbine's performance was however well-known from earlier blind tests. The scatter in the performance data of the downstream turbine at design conditions is generally observed to be larger. For 5.18 rotor diameters separation distance the $C_{P;T2}$ predictions varied within ±20%. By decreasing the separation distance to 2.77D the deviations from the measured results reduced to ±15%, while an increase in separation distance to 9.00D resulted in an even bigger scatter of ±30% in all the predictions. The scatter in the downstream turbine thrust coefficient is commonly seen to be smaller than in the power coefficient, while a tendency of underpredicting the measured thrust value is observed. Nevertheless, a significant improvement in the predictions of downstream the turbine's performance is observed compared to earlier blind test experiments, in which the scatter was more than ±100% (Pierella et al., 2013) respectively ±50% (Krogstad et al., 2014).

Comparing wake profiles behind the upstream turbine it can be concluded that both CD-adapco's IDDES computations and UU-DTU's LES simulation consistently deliver very accurate predictions of the experimentally measured mean and turbulent characteristics for all inflow conditions and separation distances. CD-adapco and UU-DTU clearly score highest in the statistical correlation coefficients for all the test cases. It seems that CD-adapco's IDDES simulations have a marginally

better resolution of flow details as reflected in very accurate predictions of the shape of the mean velocity and turbulence intensity profiles. This could be due to a better resolution of the small scales in the boundary layers of the rotor, hub and tower geometry, in which the IDDES technique takes advantage of a finer grid resolution in a RANS model. The very precise predictions of the wake shape are also confirmed in a marginally higher score in the correlation coefficients $R_{U*}$ and

$R_{k*}$, which describe correlation of the profile shape well, but are insensitive to an offset or multiplication of the data points. On the other hand, UU-DTU's simulations predicted the levels of mean velocity deficit slightly better. CD-adapco's mean velocity results have the tendency to predict a marginally too high velocity deficit, which is reflected in somewhat higher values of the mean geometrical bias $MG_{U*}$ and geometrical variance $VG_{U*}$ compared to UU-DTU's generally very precise prediction of the mean velocity levels.

The mean wake profiles are well predicted by the fully resolved k-ω SST simulations by LUT, whereas the rotor generated turbulence in the wake is clearly underpredicted. Simulations by the same group based on an actuator line approach are observed not to resolve the flow structures in sufficient detail, which is indicated by somewhat poorer averaged correlation coefficients $R_{U*}$ and $R_{k*}$ for the ACL than for the FRR approach.

CMR's wake predictions based on the k-ε turbulence model mostly manages to approximate the levels of mean velocity

deficit reasonably well; the details are however often lost due to an overprediction of turbulent diffusion. This is also the case for the k-ω simulations by Vrije, in which acceptable approximations of the mean velocity deficit for high background turbulence inflows are predicted, while the predicted turbulence distributions are observed to be too smooth. The challenges of the more complex non-uniform shear flow were resolved fairly well by most of the simulations, as most of them were able to predict a slightly skewed wake.

The discussion in the workshop disclosed that the quality of the wake predictions is dependent not only on the turbulence model, but rather a complex combination of user-dependent factors. This could be e.g. different methods of meshing, choice of turbulence parameters or force coefficients for rotor modelling. Nevertheless, this blind test also confirms that it is possible to make very accurate performance and wake flow predictions given the model and input parameters are chosen correctly.

**Acknowledgements**

The authors would like to thank the support of NOWITECH for the organisation of the workshop.

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

**Table 1.** Overview of simulation methods and parameters. Abbreviations for rotor models: *Actuator Line (ACL), Blade Element Momentum (BEM), Fully Resolved Rotor (FRR)*. Abbreviations for flow models: *Improved Delayed Detached Eddy Simulation (IDDES), Large Eddy Simulation (LES), Reynolds Averaged Navier-Stokes Simulation (RANS)*.

| | | *Simulation software* | *Rotor model* | *Airfoil data* | *Flow/Turbulence model* | *Mesh properties* | *Number of cells/nodes* | *Tunnel blockage* |
|---|---|---|---|---|---|---|---|---|
| *UU-DTU* | ▶ | EllipSys3D | ACL | Exp. DTU | LES | Cartesian | $2.9 \cdot 10^7$ cells | Yes |
| *Vrije (flow)* | ▣ | OpenFOAM | FRR | - | RANS $k\text{-}\omega$ | Hexahedral | $3.5 \cdot 10^7$ cells | Yes |
| *Vrije (forces)* | ▣ | Matlab | BEM | XFoil | - | - | - | - |
| *LUT (ACL)* | ◆ | ANSYS CFX | ACL | XFoil | RANS $k\text{-}\omega$ SST | Arbitrary | $3.7 \cdot 10^6$ nodes | Yes |
| *LUT (FRR)* | ◆ | ANSYS CFX | FRR | - | RANS $k\text{-}\omega$ SST | Structured | $4.2 \cdot 10^4$ nodes | Yes |
| *CD-adapco* | ★ | Star-CCM+ | FRR | - | IDDES $Sp.\text{-}Al.$ | Hexah./Polyh. | $2.5 \cdot 10^7$ cells | Yes |
| *CMR* | ◀ | Music | BEM | XFoil | RANS $k\text{-}\varepsilon$ | Structured | $5.0 \cdot 10^5$ nodes | Yes |

**Table 2.** Overview of turbine operating conditions, downstream turbine positions as well as wake measurement positions for the five different test cases.

| *Test case* | *Inflow* | *Inlet turbulence at position of T1* | *Tip speed ratio $\lambda_{T1}$* | *Position x/D of downstream turbine T2* | *Tip speed ratio $\lambda_{T2}$* | *Wake measurement position at x/D* |
|---|---|---|---|---|---|---|
| **A** | uniform | 0.23% | 6.0 | 5.18 | 4.5 | 2.77 |
| **B1** | uniform | 10.0% | 6.0 | 2.77 | 4.5 | - |
| **B2** | uniform | 10.0% | 6.0 | 5.18 | 4.5 | - |
| **B3** | uniform | 10.0% | 6.0 | 9.00 | 4.5 | 2.77 / 5.18 / 8.50 |
| **C** | shear | 10.1% | 6.0 | 5.18 | 4.5 | 2.77 |

**Table 3.** Numerical values of power coefficient $C_P$ and thrust coefficient $C_T$ for test case A. The downstream turbine T2 is positioned at 5.18D downstream of T1. T1 is operated at $\lambda_{T1}$=6.0 and T2 is operated at $\lambda_{T1}$=4.5 referred to the far upstream reference velocity $U_{ref}$=11.5m/s.

| | | Upstream turbine T1 | | Downstream turbine T2 | |
|---|---|---|---|---|---|
| | | $C_{P,T1}$ | $C_{T,T1}$ | $C_{P,T2}$ | $C_{T,T2}$ |
| UU-DTU | ▶ | 0.428 | 0.748 | 0.108 | 0.379 |
| Vrije | ■ | 0.457 | 0.856 | 0.244 | 0.502 |
| LUT (FRR) | ◆ | 0.468 | 0.766 | 0.171 | 0.394 |
| CD-adapco | ★ | 0.470 | 0.820 | 0.170 | 0.460 |
| CMR | ◀ | 0.433 | 0.785 | 0.158 | 0.415 |
| Experiment | ●○ | 0.462 | 0.811 | 0.145 | 0.427 |

**Table 4.** Statistical performance measures FB, NMSE, MG, VG and R of the normalized mean velocity U* normalized turbulent kinetic energy k* predictions of the five different models for test case A. The wake flow is predicted at stream-wise measurement position x/D=2.77 downstream of T1.

| | | $FB_{U*}$ | $NMSE_{U*}$ | $MG_{U*}$ | $VG_{U*}$ | $R_{U*}$ | $FB_{k*}$ | $NMSE_{k*}$ | $MG_{k*}$ | $VG_{k*}$ | $R_{k*}$ |
|---|---|---|---|---|---|---|---|---|---|---|---|
| UU-DTU | ▶ | 0.031 | 0.001 | 1.032 | 1.010 | 0.927 | -0.047 | 0.002 | 1.797 | 6.828 | 0.870 |
| Vrije | ■ | -0.081 | 0.007 | 0.897 | 1.041 | 0.895 | -0.218 | 0.048 | 0.411 | 6.038 | 0.669 |
| LUT (FRR) | ◆ | -0.009 | 0.000 | 0.980 | 1.017 | 0.877 | 0.675 | 0.515 | 1.522 | 1.879 | 0.547 |
| CD-adapco | ★ | 0.042 | 0.002 | 1.047 | 1.006 | 0.960 | -0.206 | 0.043 | 0.918 | 2.528 | 0.938 |
| CMR | ◀ | 0.000 | 0.000 | 0.988 | 1.016 | 0.886 | -1.019 | 1.404 | 0.338 | 89.922 | 0.378 |

**Table 5.** Numerical values of power coefficient $C_P$ and thrust coefficient $C_T$ for test cases $B_1$, $B_2$ and $B_3$. The downstream turbine T2 is positioned at 2.77D ($B_1$), 5.18D ($B_2$) and 9.00D ($B_3$) downstream of T1. T1 is operated at $\lambda_{T1}$=6.0 and T2 is operated at $\lambda_{T1}$=4.5 referred to the reference velocity $U_{ref}$=11.5m/s.

| | | Upstream turbine T1 | | Downstream turbine T2 at 2.77D ($B_1$) | | Downstream turbine T2 at 5.18D ($B_2$) | | Downstream turbine T2 at 9.00D ($B_3$) | |
|---|---|---|---|---|---|---|---|---|---|
| | | $C_{P,T1}$ | $C_{T,T1}$ | $C_{P,T2}$ | $C_{T,T2}$ | $C_{P,T2}$ | $C_{T,T2}$ | $C_{P,T2}$ | $C_{T,T2}$ |
| UU-DTU | ▶ | 0.447 | 0.758 | 0.115 | 0.383 | 0.152 | 0.423 | 0.192 | 0.462 |
| Vrije | ■ | 0.453 | 0.853 | 0.115 | 0.336 | 0.149 | 0.415 | 0.166 | 0.486 |
| LUT (ACL) | ◆ | 0.453 | 0.788 | 0.157 | 0.449 | 0.228 | 0.518 | 0.339 | 0.605 |
| LUT (FRR) | ◆ | 0.456 | 0.756 | - | - | 0.194 | 0.419 | - | - |
| CD-adapco | ★ | 0.470 | 0.830 | 0.130 | 0.410 | 0.170 | 0.440 | 0.230 | 0.480 |
| CMR | ◀ | 0.436 | 0.785 | 0.145 | 0.411 | 0.218 | 0.490 | 0.294 | 0.576 |
| Experiment | ●○ | 0.468 | 0.833 | 0.137 | 0.423 | 0.188 | 0.500 | 0.270 | 0.569 |

**Table 6.** Statistical performance measures FB, NMSE, MG, VG and R of the normalized mean velocity U* normalized turbulent kinetic energy k* predictions of the five different models for test case B3. The wake flow is predicted at stream-wise measurement positions x/D=2.77, 5.18 and 8.50 downstream of T1.

| | | | $FB_{U*}$ | $NMSE_{U*}$ | $MG_{U*}$ | $VG_{U*}$ | $R_{U*}$ | $FB_{k*}$ | $NMSE_{k*}$ | $MG_{k*}$ | $VG_{k*}$ | $R_{k*}$ |
|---|---|---|---|---|---|---|---|---|---|---|---|---|
| x/D = 2.77 | UU-DTU | ▶ | 0.027 | 0.001 | 1.025 | 1.002 | 0.968 | -0.329 | 0.111 | 0.671 | 1.219 | 0.911 |
| | Vrije | ■ | 0.003 | 0.000 | 1.005 | 1.002 | 0.959 | 0.222 | 0.050 | 1.239 | 1.847 | -0.008 |
| | LUT (ACL) | ◆ | -0.013 | 0.000 | 0.981 | 1.013 | 0.845 | 0.055 | 0.003 | 1.048 | 1.243 | 0.468 |
| | LUT (FRR) | ◆ | -0.009 | 0.000 | 0.988 | 1.003 | 0.949 | 0.525 | 0.296 | 1.771 | 1.539 | 0.720 |
| | CD-adapco | ★ | 0.048 | 0.002 | 1.060 | 1.006 | 0.970 | -0.007 | 0.000 | 1.035 | 1.057 | 0.912 |
| | CMR | ◀ | -0.014 | 0.000 | 0.982 | 1.007 | 0.913 | -0.771 | 0.698 | 0.404 | 2.720 | 0.417 |
| x/D = 5.18 | UU-DTU | ▶ | 0.021 | 0.000 | 1.017 | 1.002 | 0.964 | -0.203 | 0.041 | 0.794 | 1.124 | 0.850 |
| | Vrije | ■ | 0.020 | 0.000 | 1.024 | 1.003 | 0.957 | 0.047 | 0.002 | 0.988 | 1.361 | 0.371 |
| | LUT (ACL) | ◆ | -0.035 | 0.001 | 0.954 | 1.012 | 0.929 | 0.423 | 0.188 | 1.459 | 1.405 | 0.273 |
| | CD-adapco | ★ | 0.054 | 0.003 | 1.065 | 1.007 | 0.971 | -0.128 | 0.017 | 0.942 | 1.059 | 0.934 |
| | CMR | ◀ | -0.030 | 0.001 | 0.963 | 1.005 | 0.937 | -0.598 | 0.393 | 0.483 | 1.980 | 0.705 |
| x/D = 8.50 | UU-DTU | ▶ | 0.028 | 0.001 | 1.029 | 1.001 | 0.970 | -0.059 | 0.004 | 0.964 | 1.052 | 0.812 |
| | Vrije | ■ | 0.062 | 0.004 | 1.078 | 1.014 | 0.958 | -0.159 | 0.026 | 0.830 | 1.112 | 0.656 |
| | LUT (ACL) | ◆ | 0.018 | 0.000 | 1.015 | 1.001 | 0.936 | 0.706 | 0.569 | 2.095 | 1.828 | 0.594 |
| | CD-adapco | ★ | 0.116 | 0.013 | 1.143 | 1.032 | 0.962 | 0.166 | 0.028 | 1.259 | 1.130 | 0.811 |
| | CMR | ◀ | -0.040 | 0.002 | 0.957 | 1.004 | 0.955 | -0.465 | 0.228 | 0.596 | 1.410 | 0.804 |

**Table 7.** Numerical values of power coefficient $C_P$ and thrust coefficient $C_T$ for test case C. The downstream turbine T2 is positioned at 5.18D downstream of T1. T1 is operated at $\lambda_{T1}$=6.0 and T2 is operated at $\lambda_{T1}$=4.5 referred to the reference velocity $U_{ref}$=11.5m/s measured at hub height.

| | | Upstream turbine T1 | | Downstream turbine T2 | |
|---|---|---|---|---|---|
| | | $C_{P,T1}$ | $C_{T,T1}$ | $C_{P,T2}$ | $C_{T,T2}$ |
| *UU-DTU* | ▶ | 0.432 | 0.745 | 0.139 | 0.405 |
| *LUT (FRR)* | ◆ | 0.451 | 0.758 | 0.197 | 0.426 |
| *CD-adapco* | ★ | 0.460 | 0.830 | 0.170 | 0.450 |
| *CMR* | ◀ | 0.431 | 0.782 | 0.182 | 0.452 |
| *Experiment* | ●○ | 0.453 | 0.785 | 0.184 | 0.486 |

**Table 8.** Statistical performance measures FB, NMSE, MG, VG and R of the normalized mean velocity U* normalized turbulent kinetic energy k* predictions of the four different models for test case C. The wake flow is predicted at stream-wise measurement position x/D=2.77 downstream of T1.

| | | $FB_{U*}$ | $NMSE_{U*}$ | $MG_{U*}$ | $VG_{U*}$ | $R_{U*}$ | $FB_{k*}$ | $NMSE_{k*}$ | $MG_{k*}$ | $VG_{k*}$ | $R_{k*}$ |
|---|---|---|---|---|---|---|---|---|---|---|---|
| *UU-DTU* | ▶ | 0.042 | 0.002 | 1.038 | 1.003 | 0.965 | -0.246 | 0.061 | 0.684 | 1.353 | 0.866 |
| *LUT (FRR)* | ◆ | -0.005 | 0.000 | 0.986 | 1.004 | 0.952 | -0.081 | 0.007 | 0.788 | 1.475 | 0.666 |
| *CD-adapco* | ★ | 0.061 | 0.004 | 1.072 | 1.007 | 0.972 | 0.068 | 0.005 | 1.041 | 1.170 | 0.795 |
| *CMR* | ◀ | -0.002 | 0.000 | 0.993 | 1.009 | 0.898 | -0.517 | 0.286 | 0.493 | 2.161 | 0.742 |

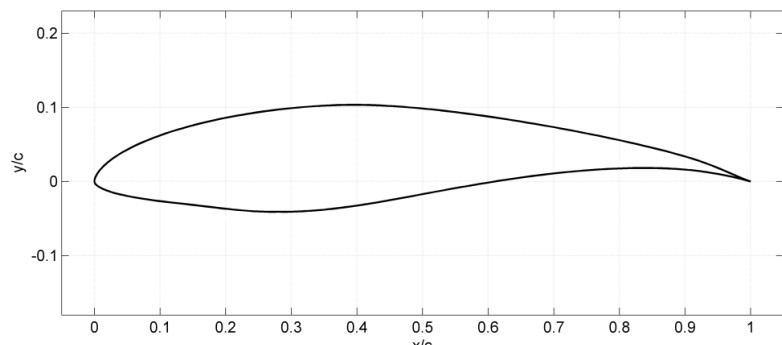

**Figure 1.** NREL S826 airfoil geometry.

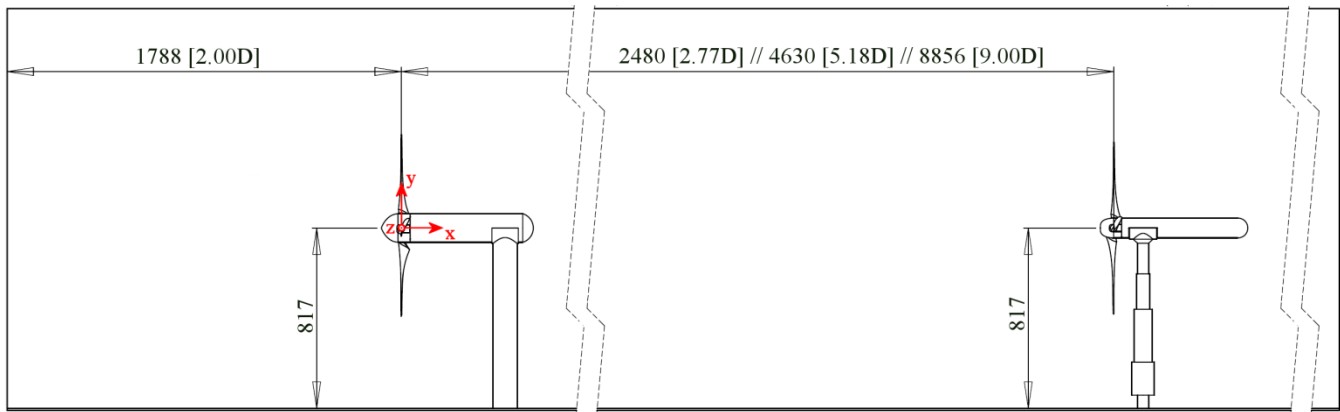

**Figure 2.** Setup of the model wind turbines in the wind tunnel and reference coordinate system.

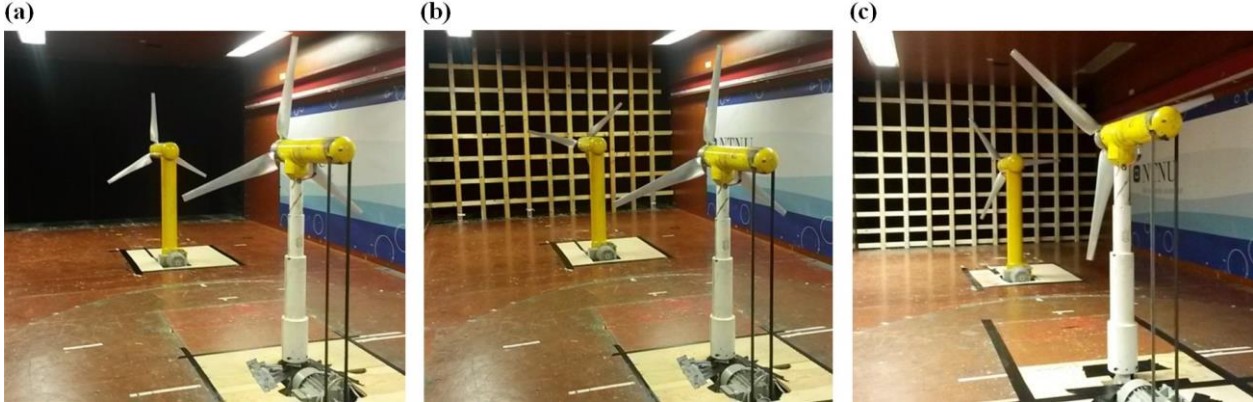

**Figure 3.** Test case A: low turbulence uniform inflow (**a**); test case B: high turbulence uniform inflow (**b**); test case C: high turbulence shear inflow (**c**).

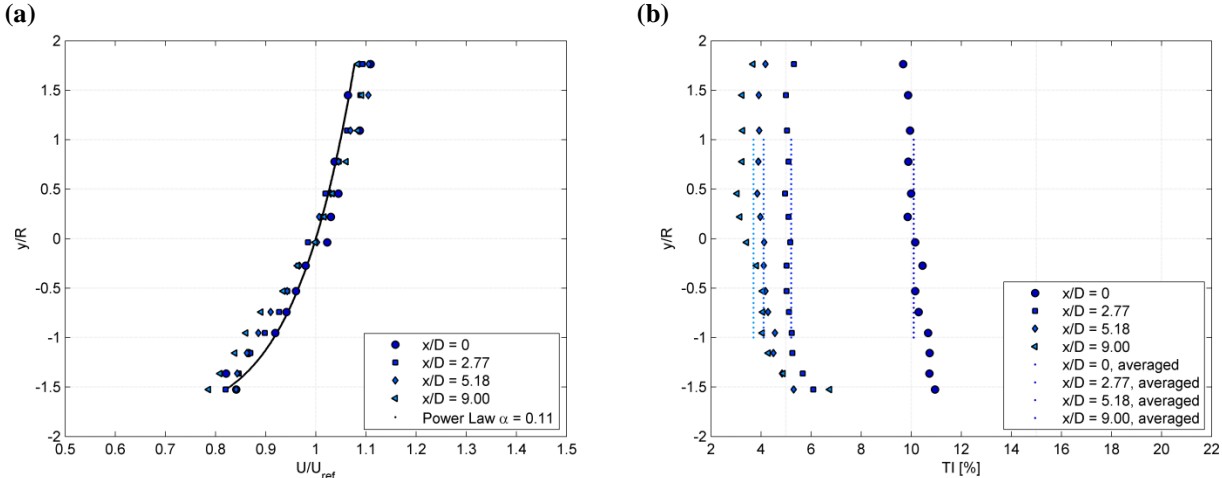

**Figure 4.** Measured and rotor-averaged values of Normalized mean velocity $U/U_{ref}$ (**a**) and Turbulence intensity TI [%] (**b**) at the position of T1 (x/D=0) and the positions of T2 (x/D=0, 2.77, 5.18, 9.00) in the empty tunnel for test case C (shear flow grid).

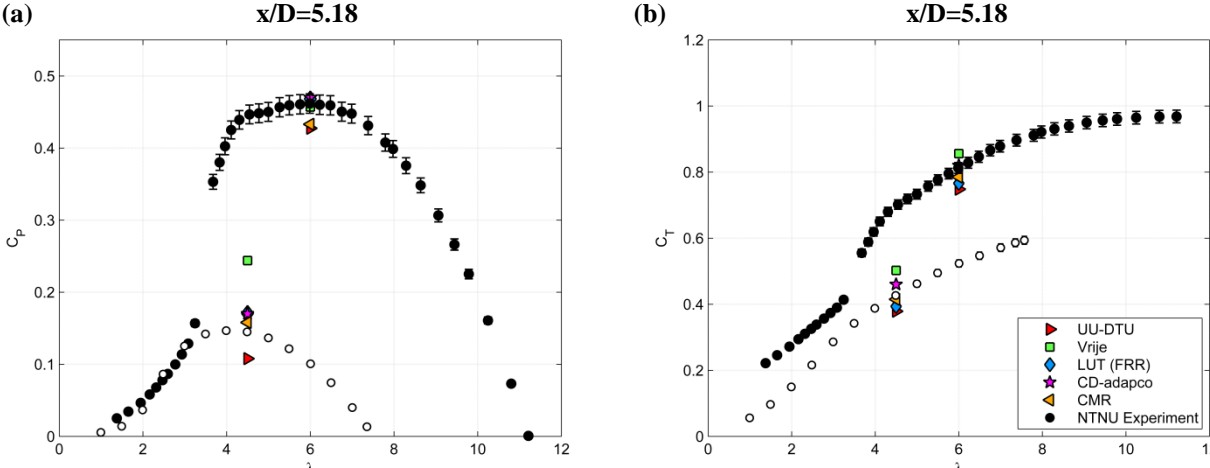

**Figure 5.** Power coefficient $C_P$ **(a)** and Thrust coefficient $C_T$ **(b)** for T1 (filled circles) and T2 (empty circles) compared for test case A. The downstream turbine T2 is positioned at x/D=5.18 downstream of T1 and the upstream turbine T1 is operated at $\lambda_{T1}$=6.0. The reference velocity is $U_{ref}$ = 11.5m/s.

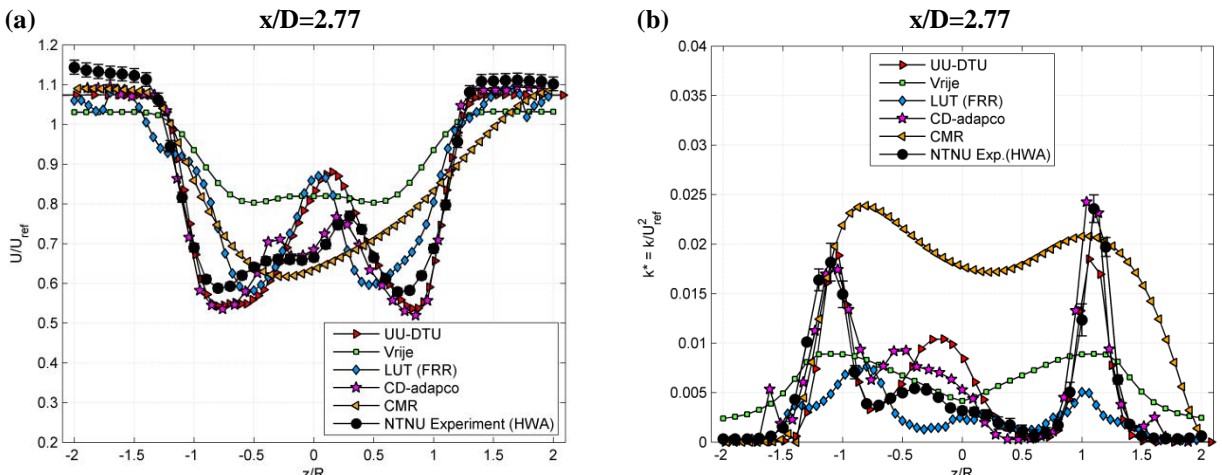

**Figure 6.** Normalized mean velocity $U/U_{ref}$ **(a)** and normalized turbulent kinetic energy $k/U_{ref}$ **(b)** in the wake x/D=2.77 behind T1 measured for test case A. The upstream turbine T1 is operated at $\lambda_{T1}$=6.0. The reference velocity is $U_{ref}$ = 11.5m/s.

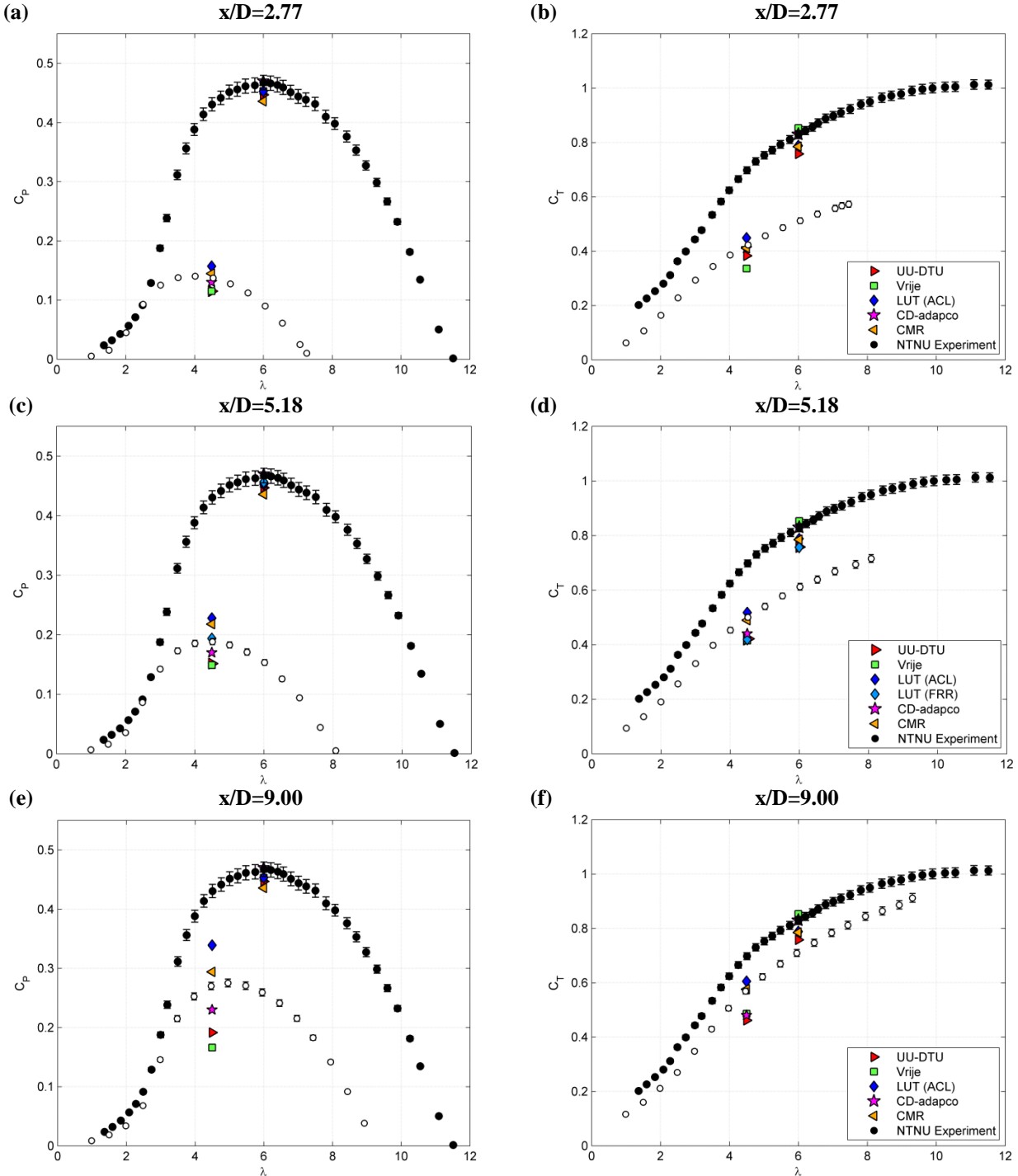

**Figure 7.** Power coefficient $C_P$ (**a, c, e**) and Thrust coefficient $C_T$ (**b, d, f**) for T1 (filled symbols) and T2 (empty circles) compared for test case $B_1$, $B_2$ and $B_3$. The downstream turbine T2 is positioned at x/D=2.77 (**a, b**), 5.18 (**c, d**) and 9.00 (**e, f**) downstream of T1. The upstream turbine T1 is operated at $\lambda_{T1}$=6.0. The reference velocity is $U_{ref}$ = 11.5m/s.

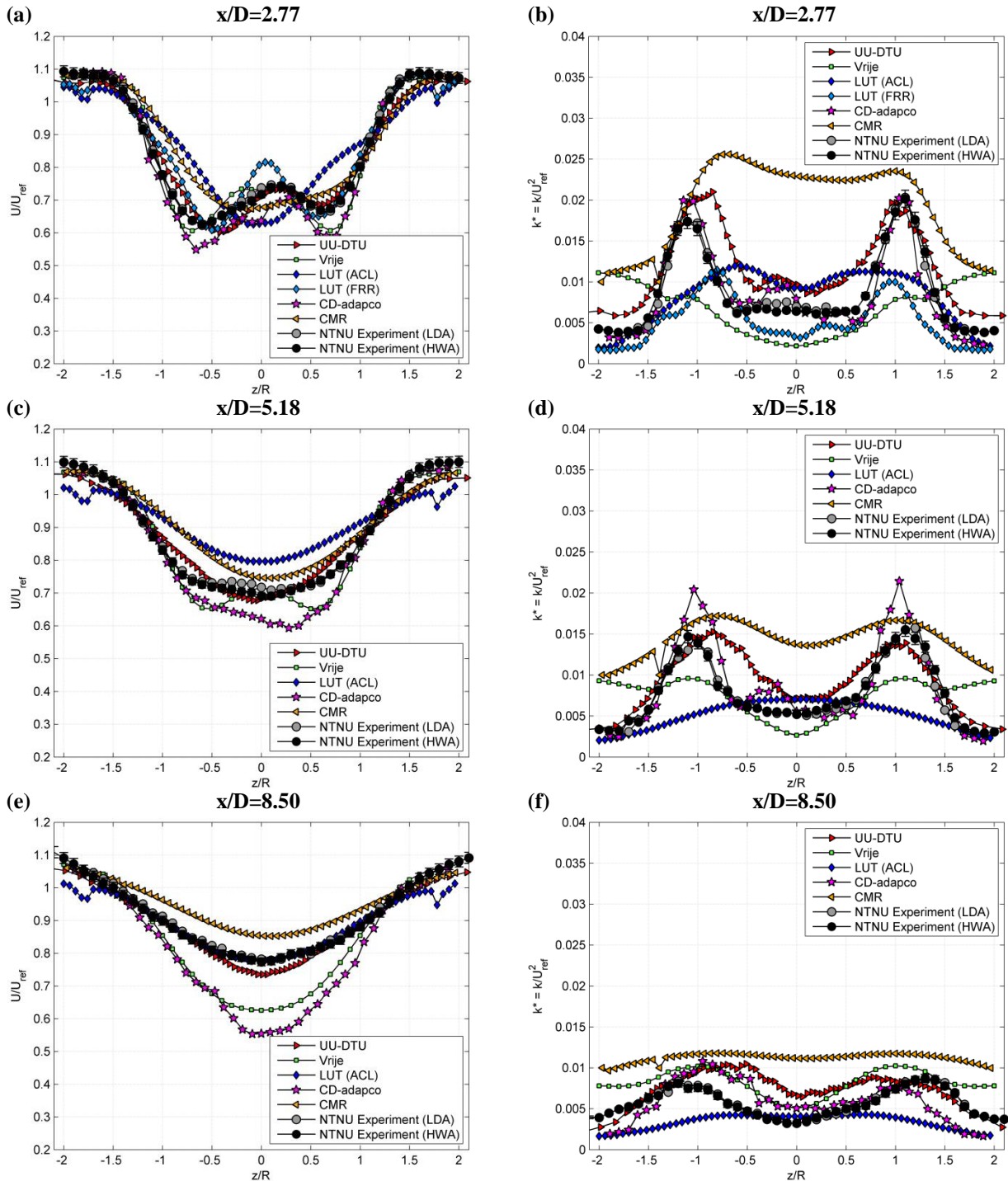

**Figure 8.** Normalized mean velocity U/U$_{ref}$ (**a, c, e**) and normalized turbulent kinetic energy k/U$^2_{ref}$ (**b, d, f**) in the wake x/D=2.77 (**a, b**), 5.18 (**c, d**) and 8.50 (**e, f**) behind T1 for test case setup B$_3$. The upstream turbine T1 is operated at $\lambda_{T1}$=6.0. The reference velocity is U$_{ref}$ = 11.5m/s.

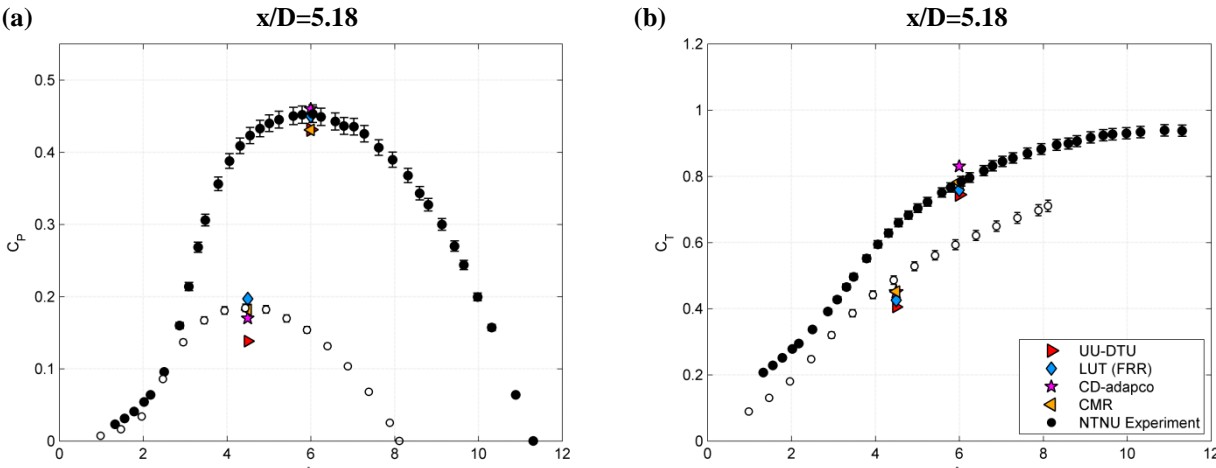

**Figure 9.** Power coefficient $C_P$ **(a)** and Thrust coefficient $C_T$ **(b)** for T1 (filled symbols) and T2 (empty circles) compared for test case C. The downstream turbine T2 is positioned at x/D=5.18 downstream of T1 and the upstream turbine T1 is operated at $\lambda_{T1}$=6.0. The reference velocity $U_{ref}$ = 11.5m/s is the velocity experienced by T1 at hub height.

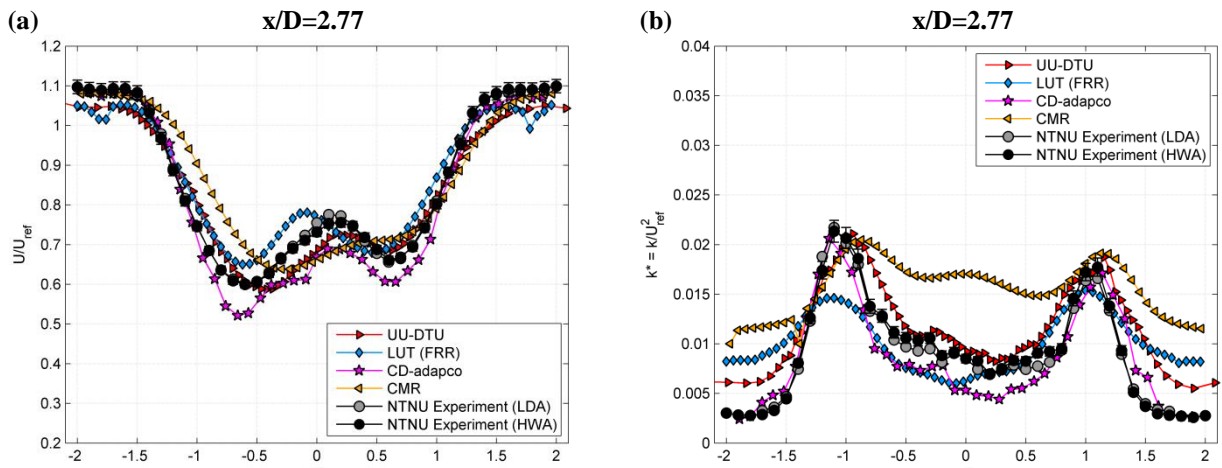

**Figure 10.** Normalized mean velocity $U/U_{ref}$ **(a)** and normalized turbulent kinetic energy $k/U_{ref}$ **(b)** in the wake x/D=2.77 behind T1 measured for test case C. The upstream turbine T1 is operated at $\lambda_{T1}$=6.0. The reference velocity $U_{ref}$ = 11.5m/s is the velocity experienced by T1 at hub height.