# Peer review of "Blind test comparison of the performance and wake flow between two in-line wind turbines exposed to different turbulent inflow conditions"

_Wind Energy Science, 2016_

## Referee Comment (RC1) · Anonymous Referee #1 · 26 Sep 2016

The manuscript deals with a blind test comparison between numerical results obtained with 5 different CFD codes and experimental results obtained in the closed-loop wind tunnel at NTNU in Trondheim. The test case is composed of 2 in-line model wind turbines exposed to different inflow conditions: homogeneous and low turbulent flow, homogeneous and high turbulent flow and, high turbulent shear flow. The comparison is based on the wind turbine performances (power and thrust) and on the flow field in the wake of the upstream wind turbine. This blind test comparison is in the continuity of several previous blind tests performed in the same facility and with the same wind turbines but with different inflow conditions and wind turbine arrangements. The main

objective is to provide database of increasing complexity in order to give the possibility to CFD developers to validate their codes on more and more challenging configurations.

Major comments:

This subject is of great interest and completely fulfills the needs expressed by numerical modelers to have access to comprehensive validation datasets with a progressive increase of complexity. It is therefore valuable to publish the outcomes of these benchmarks. On the other hand, the inherent weakness of these benchmarks is the difficulty to organize the comparison process in a synthetic way, with clearly stated outcomes based on metrics, and without losing the reader in details or technical digressions.

Consequently, I suggest to the authors to provide a table in §2.3, summarizing the properties of the different numerical tools (CFD methods, wind turbine model, mesh properties, blockage, etc).

And in order to make the discussions on results more straightforward, I suggest to use Metrics (as correlation coefficient (R), fractional bias (FB), normalized mean square error (NMSE), geometric mean (MG), geometric variance (GV), fraction with a factor of 2 (FAC 2)) to provide a synthetic comparison.

The inflow conditions are not representative of atmospheric flows. Indeed, the ratio between the integral length scale and the rotor diameter is not at all realistic. Consequently, do not mention "atmospheric" flows but "turbulent flows" in the title and in the body of text.

Minor comments:

- Remove Figs 1 and 3. Use Fig. 4(a) to describe the general set-up

- §2.2.3 change "measurement uncertainties" with "Statistical and measurement uncertainties"

- §2.4.2 : A single hot wire does not measure one velocity component but the velocity magnitude in the normal plane to the wire. Please be more precise when you explain the computations of k and the used approximations.

- How the TKE is computed for LDV measurements (at least, 2 velocity components are measured simultaneously) ? How does it compare with TKE assessed from hot wire measurement?

- §3.3: please show the velocity and turbulent profile for the inflow of test case C. Is it a linear shear?

- P18, line 10 (not shown in this report)...why do not use in this manuscript the equivalent wind speed as reference to compute the power coefficients if it gives better results? Please modify it

- §3.3 : the incoming flow is not isotropic anymore. How k is computed in the present case?

---

## Referee Comment (RC2) · Anonymous Referee #2 · 7 Oct 2016

The paper deals with the analysis of a blind test with the goal to simulate the power and thrust coefficients as well as the velocity profiles at different locations in a two wind turbine model set-up in tandem configuration all for various inflow conditions.

The paper is well written and easy to follow considering the many details mentioned in the comparison of the single experiments and simulations, respectively. Nevertheless, there are mainly two comments the authors should consider.

1. The inflow conditions do not reflect any special characteristics of atmospheric flows like gusts or intermittent distributions of velocity increments. Therefore the authors

should use the general term of turbulent inflow conditions instead atmospheric inflow conditions throughout the paper. In that case the use of turbulence intensity of sufficient like they did.

2. In figure 7 a), b), e) and f) gray full circles without error bars are used for the results of the first wind turbine. Is there a reason for that ?

In general it would be interesting to know if more detailed analyses with respect to e. g. spectra or even higher moments are planed with the data. Maybe some of the differences in the presented mean values are related to differences in the generated turbulence itself. This, of course, can and should not be discussed in this paper.

---

## Referee Comment (RC3) · Anonymous Referee #3 · 13 Oct 2016

The present study shows wind tunnel experiments using two in-line model wind turbines performed at the NTNU wind tunnel in Trondheim. The performances (power and thrust) of both turbines were examined and the velocity profiles in the first turbine's wake were measured. Experiments were conducted for different inflow conditions with increasing complexity and varying turbine distances.

Five different institutes predicted the results using numerical methods in a blind test comparison. Generally, it is of great relevance to have knowledge about the accuracy and performance of different codes. Further, as experiments and CFD simulations

both have their drawbacks, validation becomes very important and with it the need for reliable databases. The manuscript is well structured and offers a good comparison of the different results. Few aspects should be addressed, which are listed below.

Major comments:

- The introduction starts out with describing the importance of wake modeling, an overview of existing models and an introduction to CFD methods, which is well structured. Next, an overview of full-scale and wind tunnel experiments is given and the first three blind tests are described. Here, I think it is important to a) show how this blind test differs from the first ones and b) close the loop from the blind tests to CFD methods in the end. In other words, strengthening the importance of validation between CFD and experiments, knowledge of code performance and suitable datasets would improve the introduction.

Minor comments:

- Sections 2.1.2 and 2.1.3 have the same title. This should be taken care of.

- P.7, l. 31: the last sentence of the page is confusing to me, please reformulate.

- P.11, l.13: if a sentence ends with an equation, I think you should include a period (throughout the manuscript).

- Fig. 6: As in Fig 5, I think one legend is enough, so you do not have to place the legend in Fig. 6(b) over the graph.

- P.14, l.14: 'seem' instead of 'seems'

- Fig. 7/8: I think one can make it clearer that each row corresponds to one distance (and which). It does say it in the caption; however, I think this can be presented more intuitively.

- p. 15 l.19: Referencing the respective figure when you start writing about the thrust would help the reader here

- Same as above in l. 25

- You are inconsistent regarding British/American spelling in some cases, for example characteri(s/z)e, p.4, l.16 versus p.16 l.10. Please be consistent throughout the manuscript.

- P.16. l.2: so the wake measurements at 8.4D are influenced by the second turbine. Later in the manuscript, p. 17 l. 2, you mentioned additional

---

## Author Comment (AC4) · 26 Oct 2016

**Correction to Authors' reply to RC1-6:**
About the directional response of a hot-wire probe:
The correct values for the pitch factor range from **h=[1.1, 1.2]**. h=[0.1, 0.2] as stated in the above comment is a typing mistake and thus incorrect.

---

## Author Response (AR1)

Author's Response

**Blind test comparison of the performance and wake flow between two in-line wind turbines exposed to different turbulent inflow conditions**

(wes-2016-31)

Jan Bartl and Lars Sætran

**1. Point-by-point response to the reviews**

**1.1. Response to Anonymous Referee #1 (RC1)**

- **Major comment RC1-1:**
  (…) On the other hand, the inherent weakness of these benchmarks is the difficulty to organize the comparison process in a synthetic way, with clearly stated outcomes based on metrics, and without losing the reader in details or technical digressions. Consequently, I suggest to the authors to provide a table in §2.3, summarizing the properties of the different numerical tools (CFD methods, wind turbine model, mesh properties, blockage, etc)

  - *The authors' reply to RC1-1:*
    *We agree that a table summarizing the different CFD related specifications (method, turbine model, meshing resolution, blockage, turbulence model, ...) of the different participants will improve the clarity. A table will therefore be included in 2.3 in the revised version of the paper.*

- **Major comment RC1-2:**
  And in order to make the discussions on results more straightforward, I suggest to use Metrics (as correlation coefficient (R), fractional bias (FB), normalized mean square error (NMSE), geometric mean (MG), geometric variance (GV), fraction with a factor of 2 (FAC 2)) to provide a synthetic comparison

  - *The authors' reply to RC1-2:*
    *It is indeed a good idea to use statistical measures for the comparison of the results rather than only a qualitative description. The correlation of the modelled mean velocity data as well as the TKE data with the experimental data will be analysed using the metrics geometric mean (MG), geometric variance (GM), fractional bias (FB) and normalized mean square error (NMSE). We consider including a separate table at the end of each result section (3.1, 3.2, and 3.3) supported by some comments explaining correlation values.*

- **Major comment RC1-3:**
  The inflow conditions are not representative of atmospheric flows. Indeed, the ratio between the integral length scale and the rotor diameter is not at all realistic. Consequently, do not mention "atmospheric" flows but "turbulent flows" in the title and in the body of text.

  - *The authors' reply to RC1-3:*
    *Your comment, that the inflow conditions are not representative of atmospheric flows, is correct. The grid-generated turbulent flows were created to highlight effects of different turbulence levels and spatial non-uniformity in the inflow. The title will be changed according to your recommendation (if possible).*

- **Minor comment RC1-4:**
  Remove Figs 1 and 3. Use Fig. 4(a) to describe the general set-up

○ *The authors' reply to RC1-4:*
*We agree that there is a redundancy in Figures 1,3 and 4. Figure 1 is purely descriptive and will be removed. Figure 3, however, includes important information about the setup's geometry. A coordinate system (which was forgotten in this version) will be included.*

- **Minor comment RC1-5:**
§2.2.3 change "measurement uncertainties" with "Statistical and measurement uncertainties"

○ *The authors' reply to RC1-5:*
*The suggested formulation is more precise and will be changed accordingly.*

- **Minor comment RC1-6:**
§2.4.2 : A single hot wire does not measure one velocity component but the velocity magnitude in the normal plane to the wire. Please be more precise when you explain the computations of k and the used approximations.

○ *The authors' reply to RC1-6:*
*The description is indeed not completely precise as the hotwire experiences cooling from all three flow directions (partly due to the unavoidable cooling of the supporting prongs). According to Jørgensen [1971][1], the effective velocity measured by the single hot-wire is composed as follows:*
$$U_{eff}^2 = U_x^2 + k\,U_y^2 + h\,U_z^2.$$
*Therein, the yaw factor k has typical values of k=[0.1, 0.5] while the pitch factor h is typically ranging between h=[1.1, 1.2][1]. Furthermore, the calculation will be individually explained for the hot wire measurements and LDV measurements.*

- **Minor comment RC1-7:**
How the TKE is computed for LDV measurements (at least, 2 velocity components are measured simultaneously)? How does it compare with TKE assessed from hot wire measurement?

○ *The authors' reply to RC1-7:*
*That is indeed not sufficiently explained. Also for the LDV, measurements only the streamwise flow component is used to estimate the TKE. A second crosswise flow component is available, which was calculated and compared to the streamwise turbulent fluctuation, but not included in the data as shown in the paper. However, a reference to a previous Blind test experiment at the same facility is given (Figure 9 in Krogstad et al. 2014), in which the authors show that the TKE calculated from the isotropic normal stress approximation compares well to the TKE calculated from all three (measured) flow components. A similar comparison has been done for the present tests, but due to similar results it has been referred to the previous publication.*

- **Minor comment RC1-8:**
§3.3: please show the velocity and turbulent profile for the inflow of test case C. Is it a linear shear?

○ *The authors' reply to RC1-8:*
*The shear flow is described in more detail in chapter 2.1.3. It is not a linear shear. The shear flow profile can be approximated by the power law coefficient alpha=0.11 as described in chapter 2.1.3. For the mean velocity and turbulence intensity profiles it is referred to the invitational document by Sætran and Bartl (2015). As the shear flow case is a central topic of discussion, however, these profiles can, of course, also be included in the main paper.*

- **Minor comment RC1-9:**
* * *
[1] Jørgensen, F.E.: "Directional Sensitivity of Wire and Fibre-Film Probes." DISA Information No. 11. 1971

P18, line 10 (not shown in this report). . .why do not use in this manuscript the equivalent wind speed as reference to compute the power coefficients if it gives better results? Please modify it

- ○ ***The authors' reply to RC1-9:***
  *This sentence might have been misunderstood and can be clarified. The sentence shall explain that the power curve measured for test case C (shear) is matching the power curve for test case B (uniform) almost exactly if the rotor equivalent wind speed was used. This is somehow just a trivial statement as the rotor equivalent wind speed concept is defined to an equivalent rotor power with a rotor average wind speed. For a linear shear the rotor equivalent wind speed would be identical to the hub height reference wind speed, but that is not exactly the case here.*
  *For this blind test case, it was specifically defined a priori to use the reference wind speed at hub height, which is slightly different to the rotor equivalent wind speed. Where to define the reference wind speed is just a matter of definition. All modellers indeed used the hub height reference wind speed correctly and actually achieved a good match with the experimental results. All modellers predicted a lower power for test case C than in test case B, which was also measured. The reason for that is the fact that the reference wind speed at hub height is for this case not identical with the rotor equivalent wind speed.*

- **Minor comment RC1-10:**
  §3.3 : the incoming flow is not isotropic anymore. How k is computed in the present case?

  - ○ ***The authors' reply to RC1-10:***
    *The turbulent kinetic energy k for shear inflow is computed in the same way as in the previous cases for uniform inflow. It is correct that the incoming flow and thus also the turbulent kinetic energy are not completely isotropic in this case. For this reason, the 2 components wake data from LDV measurements will be further analysed for isotropy also for this case. If significant deviations from the isotropic approximations appear, these will be quantified and shown in a graph.*

**1.2. Reply to Anonymous Referee #2 (RC2):**

- **Comment RC2-1:**
  The inflow conditions do not reflect any special characteristics of atmospheric flows like gusts or intermittent distributions of velocity increments. Therefore the authors should use the general term of turbulent inflow conditions instead atmospheric inflow conditions throughout the paper. In that case the use of turbulence intensity of sufficient like they did.

  - ○ ***The authors' reply to RC2-1:***
    *Yes, that is a similar comment that referee #1 was already commenting on in RC1-3. It is correct that the inflow conditions are not representative of any atmospheric flows. The grid-generated turbulent flows were created to highlight effects of different turbulence levels and spatial non-uniformity in the inflow. "Atmospheric inflow" title will be changed to "turbulent flow" in the title and throughout the paper.*

- **Comment RC2-2:**
  In figure 7 a), b), e) and f) gray full circles without error bars are used for the results of the first wind turbine. Is there a reason for that?

  - ○ ***The authors' reply to RC2-2:***
    *As the results for the first turbine are the same for all three separation distances, they were only plotted in Figures 7 c) and 7 d). As it overfilled the plot when all data of all three separation distances were put into one plot, it was chosen to plot three separate graphs and grey out the identical results of the first*

*turbine. This was however not sufficiently explained in the text and it will be considered to fully plot the results of the first turbine also into plots 7 a), b), e) and f).*

- **Comment RC2-3:**
  In general it would be interesting to know if more detailed analyses with respect to e. g. spectra or even higher moments are planed with the data. Maybe some of the differences in the presented mean values are related to differences in the generated turbulence itself. This, of course, can and should not be discussed in this paper.

  - ○ *The authors' reply to RC2-3:*
    *This is a very good suggestion for future work or a possible next blind test. In the blind test workshop similar thoughts were articulated.*

**1.3. Reply to Anonymous Referee #3 (RC3):**

- **Major Comment RC3-1:**
  The introduction starts out with describing the importance of wake modeling, an overview of existing models and an introduction to CFD methods, which is well structured. Next, an overview of full-scale and wind tunnel experiments is given and the first three blind tests are described. Here, I think it is important to a) show how this blind test differs from the first ones and b) close the loop from the blind tests to CFD methods in the end. In other words, strengthening the importance of validation between CFD and experiments, knowledge of code performance and suitable datasets would improve the introduction.

  - ○ *The authors' reply to RC3-1:*
    *The authors agree that the introduction ends somewhat abruptly without specifically mentioning the increased complexity in the present blind test. A couple of lines describing the differences in this blind test will be added. Furthermore, it will be attempted to close the loop by stressing the importance for validation of CFD codes and thus strengthening the importance of such comparative methods.*

- **Minor Comment RC3-2:**
  Sections 2.1.2 and 2.1.3 have the same title. This should be taken care of.

  - ○ *The authors' reply to RC3-2:*
    *A mistake happened when editing the text into the WESD template. It should be "2.1.3 Inflow conditions". That will be taken care of.*

- **Minor Comment RC3-3:**
  P.7, l. 31: the last sentence of the page is confusing to me, please reformulate.

  - ○ *The authors' reply to RC3-3:*
    *The authors agree that this is a rather short and thus confusing formulation. Will be reformulated and better explained.*

- **Minor Comment RC3-4:**
  P.11, l.13: if a sentence ends with an equation, I think you should include a period (throughout the manuscript).

  - ○ *The authors' reply to RC3-4:*
    *We assume the referee is referring to punctuation marks as in this case a full stop at the end of the equation. That will be fixed in the final version.*

- **Minor Comment RC3-5:**
  Fig. 6: As in Fig 5, I think one legend is enough, so you do not have to place the legend in Fig. 6(b) over the graph.

  - *The authors' reply to RC3-5:*
    *The authors agree that one legend is enough, especially as it is hiding some data in this case (Fig. 6(b)). Will be removed, consistent with following graphs.*

- **Minor Comment RC3-6:**
  P.14, l.14: 'seem' instead of 'seems'

  - *The authors' reply to RC3-6:*
    *That will be corrected.*

- **Minor Comment RC3-7:**
  Fig. 7/8: I think one can make it clearer that each row corresponds to one distance (and which). It does say it in the caption; however, I think this can be presented more intuitively.

  - *The authors' reply to RC3-7:*
    *The authors agree. We consider labelling the plots with the separation distance, in order to be read more intuitively.*

- **Minor Comment RC3-8:**
  p. 15 l.19: Referencing the respective figure when you start writing about the thrust would help the reader here

  - *The authors' reply to RC3-8:*
    *A reference to the respective figure will be included.*

- **Minor Comment RC3-9:**
  Same as above in l. 25

  - *The authors' reply to RC3-9:*
    *A reference to the respective figure will be included.*

- **Minor Comment RC3-10:**
  You are inconsistent regarding British/American spelling in some cases, for example characteri(s/z)e, p.4, l.16 versus p.16 l.10. Please be consistent throughout the manuscript.

  - *The authors' reply to RC3-10:*
    *The final version will be proofread with special attention to BE / AE spelling.*

- **Minor Comment RC3-11:**
  P.16. l.2: so the wake measurements at 8.4D are influenced by the second turbine. Later in the manuscript, p. 17 l. 2, you mentioned additional

  - *The authors' reply to RC3-11:*
    *Unfortunately, this referee's comment seems to be incomplete. We assume the referee refers to the sentence p.17 l.10, where an additional reduction in velocity due to the presence of the turbine is mentioned. This reduction of about 5% in average is already included in the final data. The authors agree however that it could be stated more clearly. The referred sentence might be somewhat confusing and a clearer formulation will be found for the final manuscript.*

**2. List of all relevant changes**

Apart from minor changes, like spelling and formatting mistakes, the following changes have been implemented in the new version of the manuscript:

- A table summarizing the specifications of the CFD models was created (Table 1).

- The statistical performance measures FB, NMSE, MG, VG and R were calculated for all mean velocity as well as turbulent kinetic energy predictions in the wake. The performance measures are listed in tables (Table 4, 6, 8) for the different test cases. The statistical performance measures are also discussed in the result sections of the mean and turbulent wake prediction.

- The incorrect expression "atmospheric flows" was changed to "turbulent flows" in the title and main text.

- Figure 1 was removed. A coordinate system was added in Figure 3.

- The computation of the turbulent kinetic energy k from the hot-wire measurements was more thoroughly explained in chapter 2.4.2.

- The computation of the turbulent kinetic energy k from the LDV measurements was separately explained in chapter 2.4.2.

- A plot of the mean velocity and turbulence intensity over the wind tunnel height for test case C "shear inflow" has been created and added to the new version of the manuscript (Figure 4).

- The sentence about "rotor equivalent wind speed" (originally p18. l.10) was more thoroughly explained.

- The full grey circles in Figures 7 (a),(b),(e),(f) from the original manuscript have been changed to full black circles to avoid confusion.

- The introduction has been extended to include a short explanation how this Blind test experiment differs from the ones before. As suggested it has been tried to round off the introduction by highlighting the importance of validation between CFD and experiments.

- All plots have been labelled with the wake measurement location respectively the turbine separation distance to make it intuitively clearer to which row the plots refers to.

**3. Marked-up manuscript version**

A marked-up version of the manuscript is attached in the next 37 pages. All major changes, such as new table, figures and text sections are marked in yellow and commented when needed.

[revised manuscript text omitted]
 4, give a significantly poorer correlation than previously observed in the mean velocity predictions. A clear indication for the

30    off-prediction of TKE values of all models is indicated in high error values of $FB_{k*}$ and $NMSE_{k*}$. Especially the geometric variance $VG_{k*}$, which is based on a logarithmic scale, gives remarkably high deflections between $VG_{k*}=3.4$ and 29.1 for this test case. This result highlights the continuously challenging prediction of wake turbulence, especially for test cases with low background turbulence. However, fairly decent predictions are made by CD-adapco's IDDES simulation and UU-DTU's LES model, which are both approximating the profile shape well, but 
[revised manuscript text omitted]